# Tracking subjects' strategies in behavioural choice experiments at trial resolution

**Silvia Maggi[1], Rebecca M Hock[1], Martin O'Neill[1,2], Mark Buckley[3], Paula M Moran[1,4], Tobias Bast[1,4], Musa Sami[5], Mark D Humphries[1]\***

[1]School of Psychology, University of Nottingham, Nottingham, United Kingdom; [2]Department of Health & Nutritional Sciences, Atlantic Technological University, Sligo, Ireland; [3]Department of Experimental Psychology, University of Oxford, Oxford, United Kingdom; [4]Department of Neuroscience, University of Nottingham, Nottingham, United Kingdom; [5]Institute of Mental Health, University of Nottingham, Nottingham, United Kingdom

**Abstract** Investigating how, when, and what subjects learn during decision-making tasks requires tracking their choice strategies on a trial-by-trial basis. Here, we present a simple but effective probabilistic approach to tracking choice strategies at trial resolution using Bayesian evidence accumulation. We show this approach identifies both successful learning and the exploratory strategies used in decision tasks performed by humans, non-human primates, rats, and synthetic agents. Both when subjects learn and when rules change the exploratory strategies of win-stay and lose-shift, often considered complementary, are consistently used independently. Indeed, we find the use of lose-shift is strong evidence that subjects have latently learnt the salient features of a new rewarded rule. Our approach can be extended to any discrete choice strategy, and its low computational cost is ideally suited for real-time analysis and closed-loop control.

**\*For correspondence:** mark.humphries@nottingham.ac.uk

**Competing interest:** The authors declare that no competing interests exist.

## Editor's evaluation

This work describes a valuable method for indexing trial-by-trial learning and decision making strategies in animal and human behavior. The study provides compelling evidence for the validity of this new method.

## Introduction

Experiments on decision making typically take the form of discrete trials in which a subject is required to make a choice between two and more alternatives (*Packard and McGaugh, 1996*; *Yin and Knowlton, 2004*; *Behrens et al., 2007*; *Churchland et al., 2008*; *Buckley et al., 2009*; *Hanks and Summerfield, 2017*; *Izquierdo et al., 2017*; *Juavinett et al., 2018*). The subject's goal is to learn by trial and error the target rule leading to the correct choice (*Mansouri et al., 2020*), guided by feedback alone (reward and/or error signals; *Ito et al., 2015*; *Amarante et al., 2017*) or with additional predictive stimuli (such as texture or odour; *van Wingerden et al., 2010*; *Wang et al., 2019*; *Banerjee et al., 2020*). Such experimental designs can be equally applied whether the subject is primate (*Rudebeck et al., 2008*; *Buckley et al., 2009*; *Leeson et al., 2009*; *Shiner et al., 2015*), rodent (*Raposo et al., 2012*; *Brady and Floresco, 2015*; *Tait et al., 2018*; *Campagner et al., 2019*; *Harris et al., 2021*), or insect (*Giurfa and Sandoz, 2012*).

Whatever the species, neuroscience is increasingly looking to fine-grained analyses of decision-making behaviour (*Krakauer et al., 2017*; *Pereira et al., 2020*), seeking to characterise not just the variation between subjects but also a subject's variability across trials (*Smith et al., 2004*; *Roy et al., 2021*; *Ashwood et al., 2022*). When analysing the behaviour of each subject in a choice task, we ideally want to know not only when the subject has learnt the correct rule but also what the subject tried while learning. Rules correspond to particular choice strategies, like 'turn right' or 'press the cued lever'; exploratory decisions made while learning are also typically characterised as choice strategies, like 'win-stay'. Fully characterising the variation within and between individuals on choice tasks requires that we can ask of any trial: what choice strategy are they using now?

Classic approaches to analysing performance on decision-making tasks, like psychometric curves, are computed post hoc and implicitly assume a fixed strategy throughout (*Kim and Shadlen, 1999*; *Carandini and Churchland, 2013*). Choice behaviour is though inherently non-stationary: learning, by definition, requires changes to behaviour, humans and other animals switch strategies as they explore tasks, and experimenters switch target rules, introducing new rules or switching previous reward contingencies (*Birrell and Brown, 2000*; *Genovesio et al., 2005*; *Rich and Shapiro, 2007*; *Leeson et al., 2009*; *Donoso et al., 2014*; *Brady and Floresco, 2015*; *Jang et al., 2015*; *Powell and Redish, 2016*; *Bartolo and Averbeck, 2020*; *Russo et al., 2021*). Some algorithms can detect changes in choice behaviour that correspond to learning (*Smith et al., 2004*; *Suzuki and Brown, 2005*), but they lack information on what strategies subjects use to learn or whether the subject's learnt strategy matches the target rule.

We introduce a simple but effective Bayesian approach to inferring the probability of different choice strategies at trial resolution. This can be used both for inferring when subjects learn, by tracking the probability of the strategy matching the target rule, and for inferring subjects' use of exploratory strategies during learning. We show that our inference algorithm successfully tracks strategies used by synthetic agents, and then show how our approach can infer both successful learning and exploratory strategies across decision-making tasks performed by rats, non-human primates, and humans. The analysis provides general insights about learning: that win-stay and lose-shift, commonly discussed as though yoked together (e.g. *Izquierdo et al., 2017*; *Miller et al., 2017*; *Aguillon-Rodriguez et al., 2021*; *Roy et al., 2021*), are dissociable strategies; and that subjects' lose-shift choices show latent learning of new rules. Our approach is computationally efficient, could be used in real time for triggering changes to task parameters or neural stimulation, and is easily extended to more complex decision-making tasks: to aid this we provide an open source toolbox in MATLAB (GitHub, copy archived at *Humphries, 2023a*) and Python (GitHub, copy archived at *Humphries and Powell, 2023b*).

## Results

Given a subject performing a decision-making task with two or more choices, and the subject's observed history of choices up to and including trial $t$, our goal is to compute the probability $P(\text{strategy}_i(t)|\text{choices}(1 : t))$ that a particular strategy $i$ has been used on current trial $t$.

With this probability, we ask two questions about choice strategies:

1. What have the subjects learnt? By defining strategy $i$ as one of the available rules in the task, high values for $P(\text{strategy}_i(t)|\text{choices}(1 : t))$ are evidence of successfully learning that rule. This approach can be equally applied to naive subjects during training or to well-trained subjects across rule switches (e.g. to see which other rules they switch between).
2. What strategies are being tried in order to learn the rule? Subjects use choice strategies to explore the structure of the environment. For example, animals often show some combination of win-stay or lose-shift behaviour at the start of learning (*Jang et al., 2015*; *Akrami et al., 2018*; *Constantinople et al., 2019*). Potential exploratory strategies include random guessing, history dependence, and information seeking. By choosing strategy $i$ to be one of these exploratory strategies, we can track its occurrence.

### Computing strategy probabilities at trial resolution

We compute $P(\text{strategy}_i(t)|\text{choices}(1 : t))$ using Bayes theorem to provide an estimate of each probability and the uncertainty of that estimate, given the available evidence up to trial $t$:

$$\underbrace{P(\text{strategy}_i(t)|\text{choices}(1:t))}_{\text{posterior}} \propto \underbrace{P(\text{choices}(1:t)|\text{strategy}_i(t))}_{\text{likelihood}} \times \underbrace{P(\text{strategy}_i(t))}_{\text{prior}},$$

(1)

where the posterior is the observer's estimate of the probability of strategy $i$ being executed, which we want to obtain; the likelihood is the consistency of the subject's choices with the strategy we are assessing; and the prior is the initial estimate of the probability that the subject uses strategy $i$. We now define each of the likelihood, prior, and posterior distributions.

On each trial, there are two observable outcomes for a given strategy $i$: the subject's choice is either consistent with that strategy (a success, $x = 1$) or it is not (a failure, $x = 0$). The sequence of successes or failures to execute the strategy is then the history of choices [choices$(1:t)$]. As each trial in the sequence has two outcomes the likelihood $P(\text{choices}(1:t)|\text{strategy}_i(t))$ is a binomial distribution.

Setting the prior as the Beta distribution $B(\alpha_i, \beta_i)$, defined in the range [0, 1], allows us to treat the unknown probability of strategy $i$ as a random variable. The Beta distribution is the so-called conjugate prior for the binomial distribution, meaning that the posterior is also a Beta distribution, dramatically simplifying the calculations in *Equation 1*.

Indeed, so simple that updating the posterior distribution $P(\text{strategy}_i(t)|\text{choices}(1:t))$ on trial $t$ becomes, in practice, elementary arithmetic (Methods): given the parameters of the Beta distribution $(\alpha_i(t-1), \beta_i(t-1))$ for strategy $i$ on the previous trial, and that the choice on trial $t$ is either consistent $(x = 1)$ or not $(x = 0)$ with strategy $i$, then update $\alpha_i(t) \leftarrow \alpha_i(t-1) + x$ and $\beta_i(t) \leftarrow \beta_i(t-1) + (1-x)$. The initial prior is defined by the values set for $\alpha(0)$ and $\beta(0)$. This, then, uses the accumulated evidence up to trial $t$ to provide a full trial-by-trial estimate of the posterior probability of strategy $i$, in the Beta distribution $B(\alpha_i(t), \beta_i(t))$.

Using Bayes theorem in this way assumes that the parameter being estimated – here $P(\text{strategy}_i(t)|\text{choices}(1:t))$ – is stationary. But choice behaviour is often non-stationary, as when a subject switches exploratory strategies while learning, or when an experimenter changes the target rule after learning, forcing a subject to switch away from the learnt strategy. As we show further below, naive use of the Bayesian approach will fail to track these changes.

We solve the non-stationarity problem by weighting the evidence entered into the Bayesian update by its recency. We keep a running total of past successes to execute strategy $i$ up to trial $t$, exponentially weighted by how far in the past each success occurred: $s_i^*(t) = \sum_{\tau=1}^{t} \gamma^{t-\tau} x(\tau)$, where $\gamma \in (0, 1]$ is the rate of evidence decay. Similarly, we keep a running total of past failures to execute strategy $i$: $f_i^*(t) = \sum_{\tau=1}^{t} \gamma^{t-\tau}(1 - x(\tau))$. We then obtain the following algorithm for trial $t$ (Methods):

- Observe whether the choice on trial $t$ was consistent $(x(t) = 1)$ or not $(x(t) = 0)$ with the execution of strategy $i$.
- Update the running totals of evidence, decaying the prior evidence: $s_i^*(t) = \gamma s_i^*(t-1) + x(t)$; and $f_i^*(t) = \gamma f_i^*(t-1) + (1 - x(t))$.
- Update the parameters for the posterior distribution: $\alpha_i(t) \leftarrow \alpha_i(0) + s_i^*(t)$ and $\beta_i(t) \leftarrow \beta_i(0) + f_i^*(t)$.

The posterior distribution for $P(\text{strategy}_i(t)|\text{choices}(1:t))$ on trial $t$ is then the Beta distribution $B(\alpha_i(t), \beta_i(t))$. With $\gamma = 1$ we recover standard Bayesian inference; the lower $\gamma$, the stronger recent history is weighted. Throughout we use $\gamma = 0.9$ unless otherwise noted; we show below how the choice of $\gamma$ affects strategy inference.

*Figure 1* demonstrates the algorithm using example data from a naive rat performing consecutive sessions on a Y-maze task. In every trial, the rat had to choose either the left or right arm (*Figure 1a*), and received reward based on four rules that were applied in sequence: right arm, cued arm, left arm, and uncued arm. These rules switched after 10 consecutive correct trials or 11 correct out of 12 trials. *Figure 1b* plots the variables needed to infer the rat's strategy: its choices, rewards, and the cue location. We used these to compute $P(\text{strategy}_i(t)|\text{choices}(1:t))$ for the strategies corresponding to the first three rules, 'go right', 'go cued', and 'go left'. *Figure 1c* plots the resulting posterior distributions for those three strategies across three sequential trials during the learning of the 'go left' rule. These show how the rat consistently choosing the left arm updates $(\alpha, \beta)$ for $P(\text{go left})$ to shift its posterior distribution rightward, while the rat choosing not to go down the right arm when it is cued updates $(\alpha, \beta)$ for $P(\text{go cued})$ to shift its posterior distribution leftward; as a consequence, 'go left' becomes the most probable strategy.

To track changes in $P(\text{strategy}_i(t)|\text{choices}(1:t))$ across all trials we take the best estimate of the probability that strategy $i$ is being executed as the maximum a posteriori (MAP) value of $B(\alpha(t), \beta(t))$

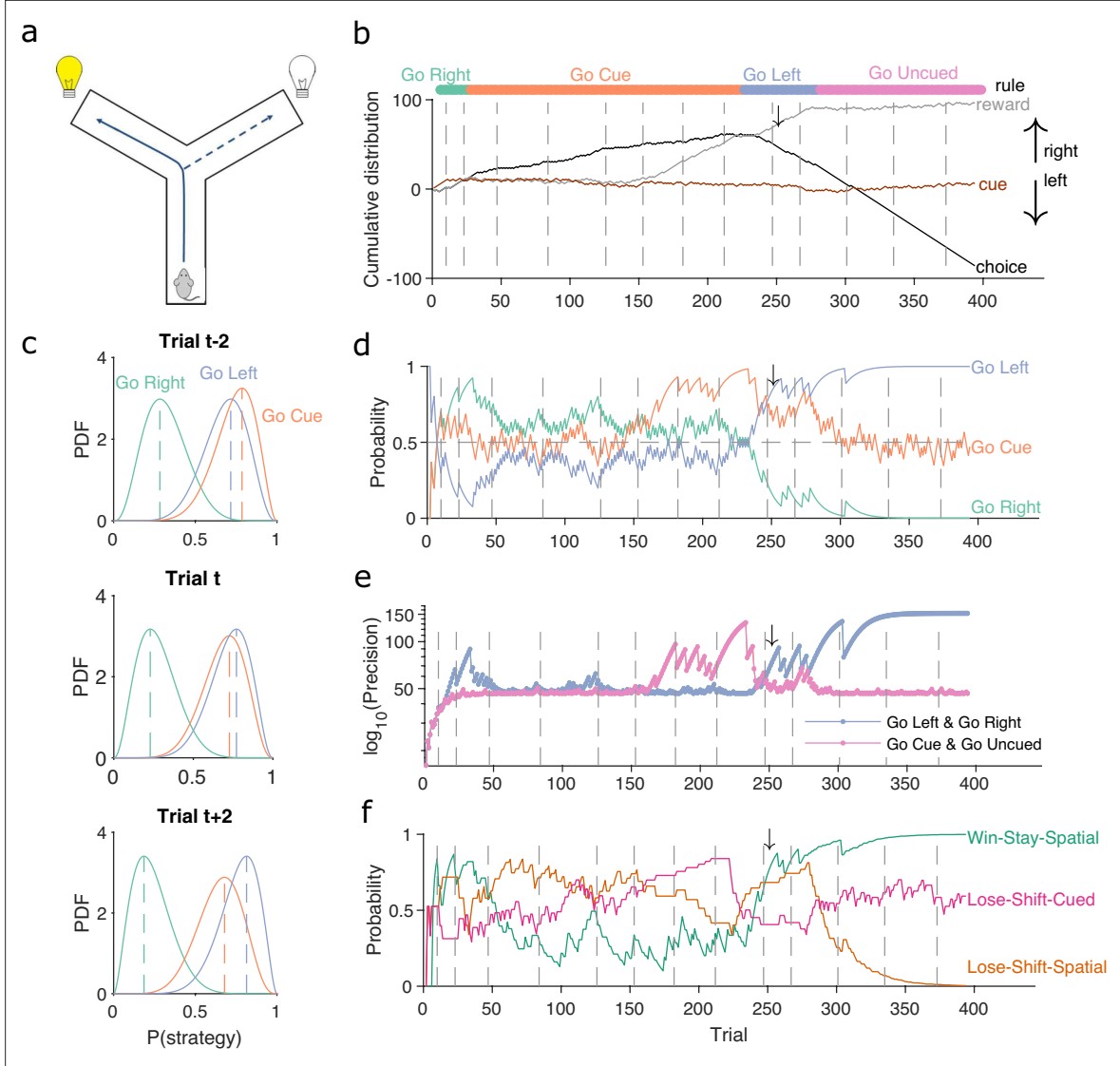

**Figure 1.** A Bayesian approach to tracking strategies. (**a**) Schematic of the Y-maze task. The rat received a reward at the arm end if it chose the correct arm according to the currently enforced rule. A light cue was randomly switched on at the end of one of the two arms on each trial. Rules were in the sequence 'go right', 'go cue', 'go left', 'go uncued', and switched after 10 consecutive correct trials or 11 out of 12. (**b**) Example rat performance on the Y-maze task. We plot performance across 14 consecutive sessions: vertical grey dashed lines indicate session separation, while coloured bars at the top give the target rule in each trial. Performance is quantified by the cumulative distributions of obtained reward (grey) and arm choice (black). We also plot the cumulative cue location (brown), to show it is randomised effectively. Choice and cue distributions increase by +1 for right and decrease by −1 for left. Small black arrow indicates the trial $t$ shown in panel c. (**c**) Example posterior distributions of three rule strategies for three sequential trials. Vertical dashed coloured lines identify the maximum a posteriori (MAP) probability estimate for each strategy. (**d**) Time-series of MAP probabilities on each trial for the three main rule strategies, for the same subject in panel b. Horizontal grey dashed line indicates chance. We omit 'go uncued' for clarity: as it is the complementary strategy of 'go cue' ($P$(go uncued) = 1 - $P$(go cued)), so it is below chance for almost all trials. (**e**) Precision (1/variance) for each trial and each of the tested strategies in panel d; note the precisions of mutually exclusive strategies (e.g. go left and go right) are identical by definition. (**f**) Time-series of MAP probabilities for three exploratory strategies. Staying or shifting is defined with respect to the choice of arm (Spatial) or the state of the light in the chosen arm (Cued) – for example if the rat initially chose the unlit arm and was unrewarded, then chose the lit arm on the next trial, this would be a successful occurrence of 'Lose-Shift-Cued'.

The online version of this article includes the following figure supplement(s) for figure 1:

**Figure supplement 1.** A stationary Bayesian approach fails to track behavioural changes.

**Figure supplement 2.** Robustness of the Bayesian approach to changes in the initial prior.

(*Figure 1c*). Plotting the MAP probabilities for each trial in *Figure 1d* shows how they capture three key dynamics of choice behaviour: first, learning when the probability the subject is using the target-rule's strategy becomes large and dominant; second, perseverance, as when learnt strategies persist after the target rule has been changed – for example, 'go right' persists as the dominant strategy for over a hundred trials after the reinforced rule has switched to 'go cued' around trial 25; and, third, switching, as when the rat switches from 'go right' to 'go cued' in the session beginning at trial 152, emphasising that choice strategy is non-stationary. Indeed, in these example data, a Bayesian approach without the decay of evidence cannot track rule learning or strategy switching (*Figure 1—figure supplement 1*). With the decay of evidence, the estimated MAP probabilities, and hence inferred changes in strategy, are robust to the choice of prior distribution (*Figure 1—figure supplement 2*).

We can also summarise the amount of evidence we have for our MAP estimate of $P(\text{strategy}_i(t)|\text{choices}(1:t))$ by computing the precision – the inverse of the variance – of the full probability distribution: the higher the precision, the more concentrated the Beta distribution is around the MAP estimate of the probability of strategy $i$ (*Figure 1e*).

The choice of scalar summaries (MAP and precision) here was made to facilitate ease of plotting (*Figure 1* onwards), quantifying (*Figure 2* onwards), and combining across subjects (*Figure 3* onwards) the changes to the posteriors $P(\text{strategy}_i(t)|\text{choices}(1:t))$. As we show below, these scalar summaries were sufficient to reveal striking behavioural changes during learning and exploration; but we emphasise that the full posterior is calculated for each trial, offering rich potential for further applications of this algorithm.

The above focused on strategies corresponding to the target rules to learn. We can equally define models for strategies that are explorative, to track the strategies the subject engages while learning the rule. For example, a subject's strategy of win-stay based on their previous choice would be: count a success if trial $t - 1$ was correct and the subject made the same choice on trial $t$; count a failure if trial $t - 1$ was correct and the subject made a different choice on trial $t$. In *Figure 1f*, we plot MAP probabilities for three such explorative strategies, win-stay and lose-shift for the prior choice of left or right arm, and lose-shift based on the prior cue position. This demonstrates when exploratory strategies are used to guide behaviour; for example, of these strategies, lose-shift in response to the choice of arm begins to dominate from around trial 50–150 (see *Figure 1f*).

Throughout we distinguish rule strategies, which correspond to one of the reinforced rules available in the task, from exploratory strategies, which are all other possible choice strategies given task information. The choice of strategies is down to the user (Methods).

## Robust tracking of strategies in synthetic data

We turn now to three questions about the algorithm and its use: first, why do we need to decay evidence? Second, how does the choice of decay rate $\gamma$ affect our inferences of $P(\text{strategy}_i(t)|\text{choices}(1:t))$? And, third, what happens if our chosen strategies do not include a true strategy? We show that decaying evidence is necessary to track strategy changes; that there is a range of $\gamma$ over which performance is robust; and that the absence of a true strategy sets upper bounds on the algorithm's estimate of $P(\text{strategy}_i(t)|\text{choices}(1:t))$.

First, to demonstrate the need to decay evidence, we simulated an agent doing a two-alternative forced-choice task. The agent was rewarded when it chose the randomly cued option (brown curve in *Figure 2a*). The agent switched its strategy every 100 trials, across five strategies, with only the fourth strategy corresponding to the rewarded rule (*Figure 2a*). This set of implanted strategies included a range of options examined in behavioural studies, often individually, and we used many consecutive trials with the same strategy to explore the full effects of the decay rate $\gamma$. We inferred $P(\text{strategy}_i(t)|\text{choices}(1:t))$ for eight strategies in total, including the five implanted ones, using the agent's choice data. *Figure 2* shows how the algorithm's MAP estimate of $P(\text{strategy}_i(t)|\text{choices}(1:t))$ (panel b) and its precision (panel c) tracked these strategy switches when using a decay rate of $\gamma = 0.9$. *Figure 2—figure supplement 1* plots every strategy's MAP estimates at all tested decay rates.

We used a simple decision rule to quantify how the decay rate $\gamma$ affected the algorithm's tracking of strategies (*Figure 2d*). This rule selected the most likely strategy being executed at trial $t$ by choosing strategy $i$ with the highest MAP estimate of $P(\text{strategy}_i(t)|\text{choices}(1:t))$, resolving ties by choosing the strategy with the maximum precision (*Figure 2d*).

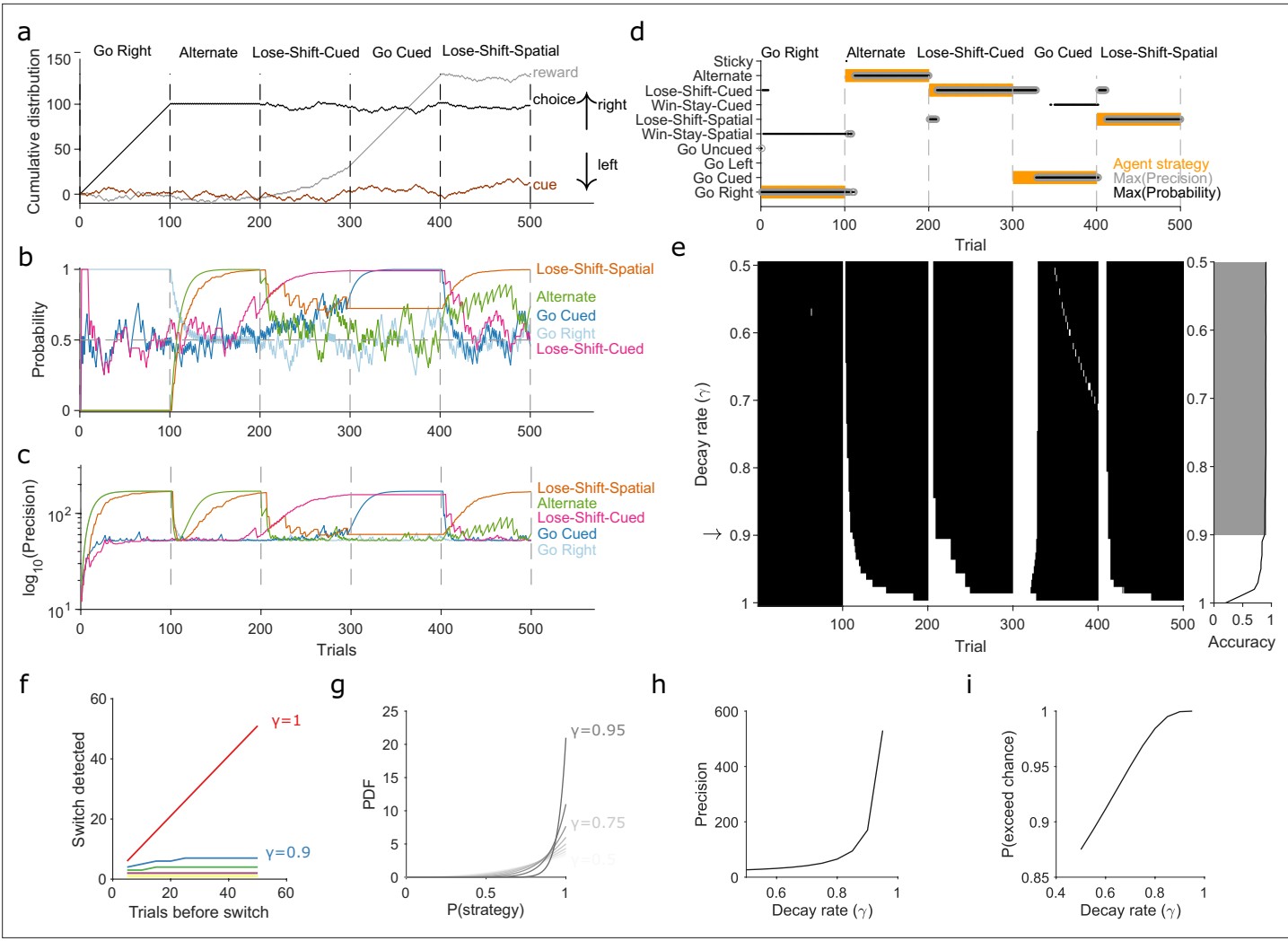

**Figure 2.** Robust tracking of strategies in synthetic data. (**a**) Cumulative distribution of raw behavioural data for the synthetic agent, performing a two-alternative task of choosing the randomly cued option. Curves are as in *Figure 1b*. Vertical dashed lines indicate strategy switches by the agent. (**b**) Maximum a posteriori (MAP) values of $P(\text{strategy}_i(t)|\text{choices}(1:t))$ for the five implanted strategies across trials, for $\gamma = 0.9$. Chance is 0.5. (**c**) Precision for the five implanted strategies. (**d**) A decision rule to quantify the effect of $\gamma$. For every tested strategy, listed on the left, we plot for each trial the strategies with the maximum MAP probability (black dots) and maximum precision (grey circles) across all strategies, and the agent's actual strategy (orange bars). Maximising both the MAP probability and precision uniquely identifies one tested strategy, which matches the agent's strategy. (**e**) Algorithm performance across evidence decay rates. Left: successful inference (black) of the agent's strategy using the decision rule. Right: proportion of trials with the correct detected strategy; grey shading shows $\gamma$ values within the top 5%. Standard Bayesian inference is $\gamma = 1$. (**f**) Number of trials until a strategy switch is detected, as a function of the number of trials using the first strategy. One line per decay rate ($\gamma$). Detection was the first trial for which the algorithm's MAP probability estimate of the new strategy was greater than the MAP probability estimate of the first strategy. (**g**) The decay rate ($\gamma$) sets an asymptotic limit on the posterior distribution of $P(\text{strategy}_i(t)|\text{choices}(1:t))$. We plot the posterior distribution here for an infinite run of successes in performing strategy $i$ at a range of $\gamma \in [0.5, 0.95]$. (**h**) The asymptotic limits on the precision of the posterior distributions set by the choice of decay rate ($\gamma$). (**i**) The asymptotic limits on the probability that $P(\text{strategy}_i(t)|\text{choices}(1:t))$ exceeds chance (p = 0.5) in a two-choice task.

The online version of this article includes the following figure supplement(s) for figure 2:

**Figure supplement 1.** Maximum a posteriori (MAP) probabilities for agent strategies across changes in decay rate.

**Figure supplement 2.** Slower evidence decay tracks gradual switches in strategy more accurately and robustly.

**Figure supplement 3.** Slower evidence decay gives lower variability when tracking stochastic use of a strategy.

**Figure supplement 4.** Algorithm performance for tracking a new conditional strategy.

**Figure supplement 5.** Limits on the probability of the tested strategy matching a true strategy used by an agent during exploration.

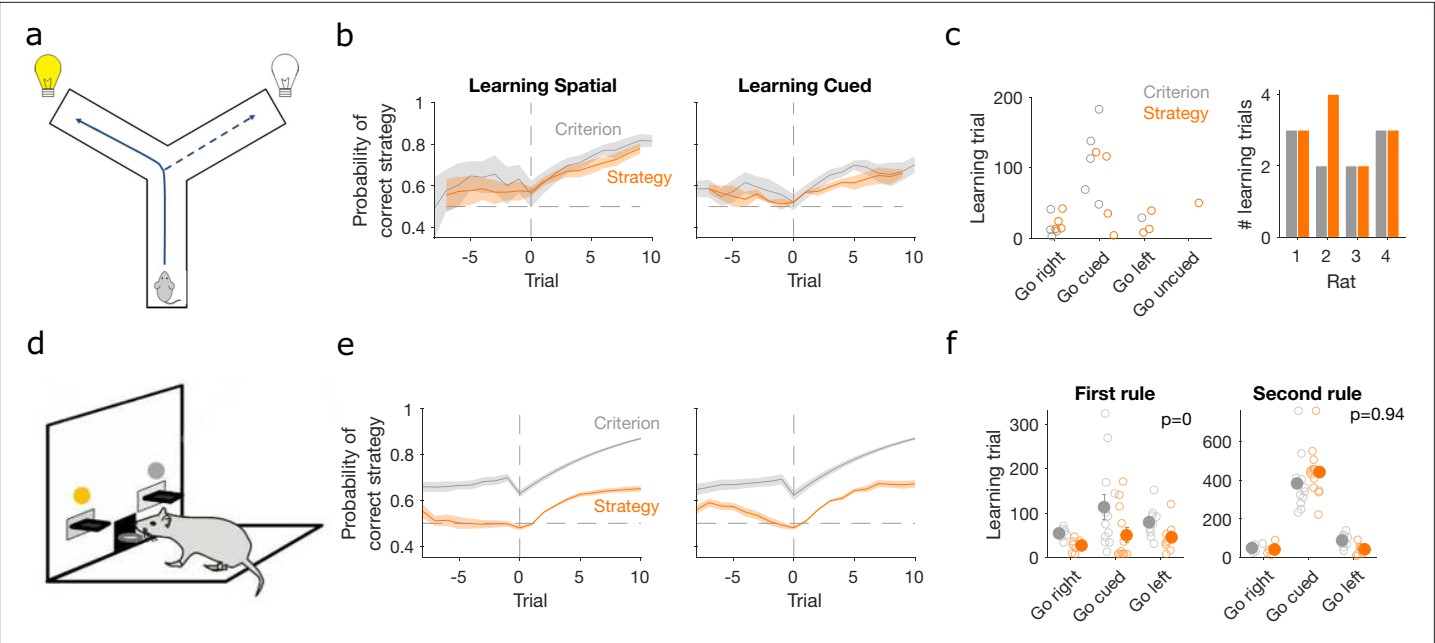

**Figure 3.** Detecting learning. (**a**) Schematic of Y-maze task, from *Figure 1a*. (**b**) Maximum a posteriori (MAP) estimate of $P(\text{strategy}_i(t)|\text{choices}(1:t))$ for the correct strategy for the spatial (left panel) and cued (right panel) rules on the Y-maze, aligned to the learning trial for each subject (trial 0, vertical dashed line). Learning trials are defined by the experimenters' original criterion (grey) or by our strategy criterion (orange). Curves represent means (bold line) ± standard error of the mean (SEM; shading) for $N$ sessions meeting each criterion across the four rats (Spatial rule: original criterion $N = 5$, strategy criterion $N = 7$; Cued rules: original criterion $N = 5$, strategy criterion $N = 5$). Horizontal dashed line is chance. (**c**) Comparison between learning trials in the Y-maze task identified using the original criterion or our strategy criterion. Left: the identified learning trials for each rule. Trial zero was the first trial with that rule enforced, and rules are labelled in the order they were presented to the rats. Right: number of identified learning trials per rat. (**d**) Schematic of the lever-press task. Rats experienced both spatial (e.g. choose left lever) and cued (e.g. choose lit lever) rules. (**e**) As for panel b, for the lever-press task. $N = 28$ rats. (**f**) Comparing the learning trials identified in the lever-press task by the original criterion or our strategy criterion for the first (left) and second (right) rule learnt by each rat. Solid symbols show mean and SEM; open symbols show individuals' learning trials. p-values are from a Wilcoxon signed rank test between the learning trials identified by the original and by the strategy criterion ($N = 28$ rats). For the first rule, the learning trials based on the original and strategy criterion are 88.9 ± 12.3 and 59 ± 11.3 (mean ± SEM), respectively. For the second rule, the learning trials based on the original and strategy criterion are 224.3 ± 33 and 272 ± 33 (mean ± SEM), respectively. Note the second rule's y-axis scale is a factor of 2 larger than for the first rule, because of the time taken to learn the cued rule second.

The online version of this article includes the following figure supplement(s) for figure 3:

**Figure supplement 1.** Performance of further learning criteria.

*Figure 2e* plots the trials on which the algorithm was correct using this decision rule. We see standard Bayesian inference ($\gamma = 1$) could not detect any switches in strategy. But with $\gamma \approx 0.9$ or less, the algorithm detected switches in 10 trials or fewer, and tracked the correct strategy in more than 88% of trials (grey shading in *Figure 2e*).

We then used extensive simulations to quantify how the number of trials $T$ for which a first strategy is used affects the algorithm's ability to track an abrupt switch to a second strategy (Methods). Standard Bayesian inference scaled linearly, taking another $T$ trials to detect the switch (*Figure 2f*, $\gamma = 1$). Using evidence decay, our algorithm detects abrupt switches orders-of-magnitude faster, and at a rate that rapidly plateaued with increasing $T$ (*Figure 2f*). Decaying evidence thus ensures that how long a first strategy was used little impacts the algorithm's ability to track a switch to a new strategy. And while stronger evidence decay (smaller $\gamma$) increased the speed at which the abrupt switch was detected, even for $\gamma = 0.9$ the worst-case performance was 10 trials (*Figure 2f*). Collectively, these results show how decaying evidence is necessary to detect switches in strategy, and that the algorithm can rapidly track abrupt switches in strategy for a wide range of $\gamma$.

We could then turn to our second question: If the decay of evidence is essential, how rapid should it be? Choosing $\gamma < 1$ defines asymptotic limits on the posterior distribution because it limits the values that $\alpha$ and $\beta$ can take when there is a long run of successes or failures to execute a strategy (Methods, *Equations 15 and 16*). These limits do not stop the algorithm correctly estimating the

MAP value of $P(\text{strategy}_i(t)|\text{choices}(1:t))$; for example, for a long run of successes, the MAP estimate of $P(\text{strategy}_i(t)|\text{choices}(1:t))$ will always be 1 (*Figure 2g*). But these limits do mean that, in contrast to standard Bayesian inference, rapid decay $\gamma \ll 1$ places bounds on how concentrated the posterior distribution of $P(\text{strategy}_i(t)|\text{choices}(1:t))$ can get around its maximum value (*Figure 2g–i*).

The main consequence of these bounds is that slower decay of evidence makes the algorithm more robust to noise. We see this robustness in two crucial uses of the algorithm, tracking gradual switches between two strategies and tracking the stochastic use of a strategy. In *Figure 2—figure supplement 2*, we show that for gradual switches slowly decaying evidence ($\gamma \sim 0.9$) correctly identifies when the new strategy has become dominant, and consistently tracks its use thereafter; by contrast, rapidly decaying evidence ($\gamma \ll 1$) prematurely identifies when the new strategy has become dominant and inconsistently tracks it use thereafter. In *Figure 2—figure supplement 3*, we show that for the stochastic use of a strategy $i$ with some probability $p(\text{use})$, the average MAP estimate $P(\text{strategy}_i(t)|\text{choices}(1:t))$ will converge to $p(\text{use})$; but we also show that the more rapidly evidence is decayed the greater the trial-to-trial variation in $P(\text{strategy}_i(t)|\text{choices}(1:t))$. Choosing the decay rate $\gamma$ is thus a trade-off between the speed (*Figure 2f*) of tracking changes in strategy and the accuracy of that tracking (*Figure 2g*, *Figure 2—figure supplement 2*, *Figure 2—figure supplement 3*). Our use of $\gamma = 0.9$ here is thus motivated by favouring robust yet still rapid tracking.

Tracking exploratory strategies presents a further challenge, our third question above: The set of exploratory strategies we test may not include a true strategy used by a subject. Exploratory strategies come in two forms: like rule strategies they can be unconditional, available for use at every decision, such as repeating a previous choice; but unlike rule strategies they can be conditional, only available for use when their condition or conditions or met, such as win-stay. We first confirmed that our algorithm could rapidly track the use of conditional strategies (*Figure 2—figure supplement 4*). We then sought to place bounds on how the inference algorithm behaves if a true exploratory strategy is missing.

What then would the algorithm's MAP estimate of $P(\text{strategy}_i(t)|\text{choices}(1:t))$ be for any kind of strategy $i$ that is not the true exploratory strategy? The general answer to this is the probability $p(\text{match})$ that the choice made under strategy $i$ matches the choice made under the true strategy. By definition $p(\text{match}) < 1$, otherwise strategy $i$ and the true strategy are indistinguishable to the observer. What we are interested in is how close $p(\text{match})$ could grow to 1, and thus how hard it would be to tell when we do not have the true strategy. Exhaustively enumerating the possible values of $p(\text{match})$ for a task with $n$ choices revealed three bounds (we derived these from the last column in *Figure 2—figure supplement 5*; see Methods for details of enumeration). First, if the algorithm estimates the MAP probability of all tested exploratory strategies to be $\leq 1 - 1/n$ then the true strategy or strategies are missing. Second, an MAP probability $> 1 - 1/n$ for a tested exploratory strategy is evidence for at least a partial match to the true strategy (e.g. in *Figure 2b*, when the true strategy adopted by the agent was 'lose-shift-spatial' the tested strategy of 'alternate' had an MAP probability of around 0.75, because both strategies will change choice on the trial after a loss, thus matching on all post-loss trials but not all post-reward trials). Indeed, our preferred way of interpreting exploratory strategies is that they provide evidence about what task features, such as obtaining reward (or not), and the subject's response to them, such as staying or shifting, are driving exploration; the algorithm's estimate of $P(\text{strategy}_i(t)|\text{choices}(1:t))$ is thus a read-out of the evidence. Third, we find that an MAP probability approaching 1 is strong evidence that the tested strategy is, or is equivalent to, the true strategy.

Collectively these results show that our approach, combining Bayesian inference with a rate of forgetting evidence used in that inference, can track non-stationary probabilities of strategy use at trial resolution. While the forgetting rate is essential (*Figure 2a–f*), the effects it has on the behaviour of the posterior distribution (*Figure 2g–i*) and on the ability to correctly track switches in strategy (*Figure 2—figure supplement 2* and *Figure 2—figure supplement 3*), give us a principled range of forgetting rates to consider, and over which the algorithm's performance is robust. With these insights to hand, we now apply our inference algorithm to uncover the choice strategies used in tasks performed by rats, humans, and monkeys.

## Trial-resolution inference reveals evidence of earlier learning

Classic approaches to detecting learning depend on arbitrary criteria based on counts of correct trials (e.g. *Birrell and Brown, 2000*; *Boulougouris et al., 2007*; *Floresco et al., 2008*; *Leeson et al., 2009*;

*Peyrache et al., 2009*; *Brady and Floresco, 2015*). Such criteria are typically conservative, requiring long sequences or high percentages of correct trials to give unambiguous evidence of learning. We thus asked whether our trial-resolution algorithm could provide a more principled approach to detecting learning.

To investigate this question, we used data from rats learning cross-modal rule-switch tasks. We compared data from two conceptually similar tasks, the first involving the choice of one of two arms in a Y-maze (*Figure 1a*, data from *Peyrache et al., 2009*) and the other the choice of one of two levers in an operant box (*Figure 3d*; task design from *Brady and Floresco, 2015*). In both tasks, the rat had to learn at least one spatial rule based on the position of the choice option (arm or lever) and at least one cue-driven rule, based on the position of a randomly illuminated cue light. As noted above, the Y-maze task contained a sequence of up to four rules, two spatial (left/right) and two cue-driven. In the data available to us, three rats experienced all four rules and one rat experienced the first two rules, giving a total of seven switches from spatial to cue-driven rules and three switches from cue-driven to spatial rules. In the lever-press task, rats were trained on one spatial and one cue-driven rule: 16 rats experienced a switch from a spatial to a cue-driven rule, and 12 experienced the switch from a cue-driven to a spatial rule (Methods).

Classic definitions of learning the target rule were originally used for both tasks. For the Y-maze study, learning was defined post hoc by three consecutive correct trials followed by 80% correct until the end of the session, with the first of the three trials then labelled as the learning trial (*Peyrache et al., 2009*). For the lever-press task, learning was defined on-line as 10 consecutive correct trials, after which the session ended (*Brady and Floresco, 2015*) – the first of these ten we label as the learning trial. To assess how conservative these criteria were, we computed the probability $P(\text{strategy}_i(t)|\text{choices}(1:t))$ for the strategy $i$ corresponding to the target rule and found it was consistently above chance before these classically defined learning trials for both the Y-maze (*Figure 3b*, 'Criterion') and lever-press tasks (*Figure 3e*, 'Criterion'). This suggests rules were learnt notably earlier than detected.

We thus tested a sequence-based criterion using $P(\text{strategy}_i(t)|\text{choices}(1:t))$ to identify the learning trial: we found the first trial at which the MAP probability for the target-rule's strategy remained above chance (0.5) until the end of the session (Methods). Using this criterion, the change in probability around the learning trial showed clearer steps from chance to above-chance performance in both tasks (*Figure 3b, e*, 'Strategy').

In the Y-maze task, this strategy-based criterion identified that learning occurred 12 times across the four rats, compared to 10 times using the original criterion (*Figure 3c*). In particular, unlike the original criterion, the strategy criterion suggested that all three rats which experienced the 'go left' rule learnt it, and one rat learnt the 'go uncued' rule.

For the lever-press task, we could compare the learning trials identified by both approaches because every rat was trained to the original criterion. The strategy criterion robustly identified learning earlier, on average 30 ± 5.8 trials earlier, during learning of the first target rule (*Figure 3f*, left). In contrast, both criteria identified similar learning trials during the second target rule each rat experienced (*Figure 3f*, right). They thus both agreed that learning the cued rule after a spatial rule took longer than learning a spatial rule after the cued rule (*Figure 3f*; a two-factor analysis of variance on the strategy criterion showing significant effects of rule type [$F = 74.73$, p < 0.001], rule order [$F = 45.36$, p < 0.001], and their interaction [$F = 54.74$, p < 0.001]).

The strategy-based criterion was analogous to classical criteria for learning as it used only a scalar estimate of performance. We thus explored two other criteria for learning (Methods) that also incorporated the uncertainty in the estimate of $P(\text{strategy}_i(t)|\text{choices}(1:t))$. The first extended the sequence-based definition to also require that the precision of the target-rule's strategy was greater than all other tested rule strategies, indicating greater evidence for that strategy. The second defined the learning trial as the first at which the probability the posterior distribution $P(\text{strategy}_i(t)|\text{choices}(1:t))$ did not contain chance was above some high threshold (here 0.95) until the end of the session. We found, as expected, that learning took longer to detect with these increasingly stringent criteria for the level of uncertainty (*Figure 3—figure supplement 1*). Indeed, the full posterior criterion seemed to identify expert performance in both tasks (*Figure 3—figure supplement 1a and c*), so could itself be a useful criterion for making rule switches. Consequently, for the lever-press task, both these more stringent criteria did not identify learning markedly earlier than classic sequence criterion (*Figure 3—figure*

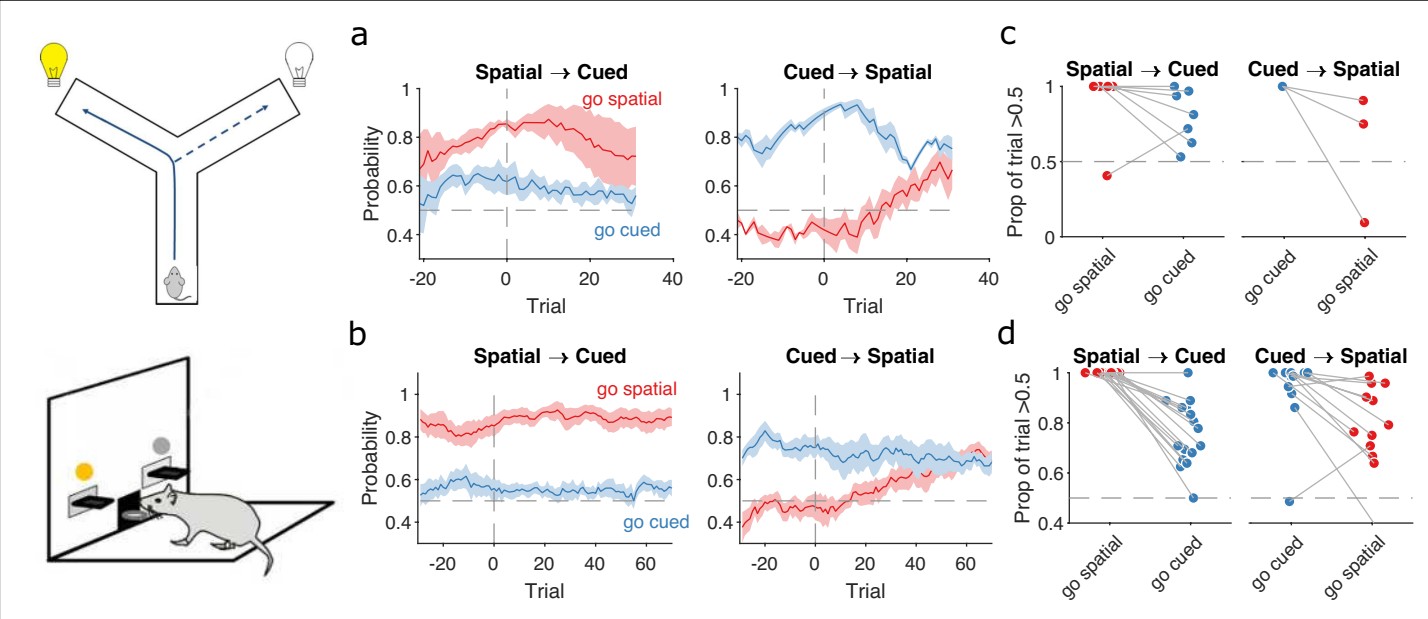

**Figure 4.** Detecting responses to task changes. (**a**) Responses to rule switches in the Y-maze task. The left panel plots the maximum a posteriori (MAP) estimates of $P(\text{strategy}_i(t)|\text{choices}(1:t))$ for the target spatial and cue strategies aligned to the switch from a spatial to a cued rule ($N = 7$ sessions, from four rats). The right panel plots the MAP estimates aligned to the switch from a cued to a spatial rule ($N = 3$ sessions, from three rats). For both panels, curves plot means ± standard error of the mean (SEM; shading) across sessions, aligned to the rule-switch trial (vertical dashed line). The horizontal dashed line indicates chance. (**b**) Same as panel a, but for the lever-press task. $N = 16$ rats for spatial → cued; $N = 12$ cued → spatial. (**c**) Response to each rule-switch in the Y-maze task. Each dot shows the proportion of trials after a rule switch in which the labelled strategy was above chance. Proportions calculated for a window of 30 trials after the rule switch. Lines join datapoints from the same switch occurrence. (**d**) Same as panel c, for the lever-press task. Proportions calculated for a window of 60 trials after the rule switch, to take advantage of the longer sessions in this task.

*supplement 1d*). Nonetheless, these more stringent criterion still showed that animals were significantly slower to learn when switching from a spatial to a cued rule than vice versa (*Figure 3—figure supplement 1d*).

### Flexibility in responses to rule changes depends on rule type

Learning a target rule is one of two key events in decision-making tasks. The other is responding to a change in the target rule. Changes in behaviour around rule switches are useful to understand the flexibility of behaviour, and the relative dominance of one mode of responding (e.g. spatial) over another (e.g. cue-driven) (*Birrell and Brown, 2000*; *Floresco et al., 2008*; *Buckley et al., 2009*; *Wang et al., 2019*). To show how our approach can quantify behavioural flexibility, we thus used our inference algorithm to examine trial-resolution changes in choice strategy around rule-switches in the same Y-maze and lever-press tasks.

In both tasks, rats which had been training on a spatial rule continued to have a high probability of executing a spatial strategy on each trial long after the rule had switched to a cue-based target rule (*Figure 4a, b*, left panels). Thus, rats showed little behavioural response during switches from spatial to cued rules.

The switch from a cued to a spatial rule in both tasks evoked a considerably faster change in strategy. Over the same time window as the spatial → cued switch, while the probability of selecting the cued strategy declined slowly, the probability of selecting the now-correct spatial strategy increased above chance within 20 trials of the rule switch (*Figure 4a, b*, right panels).

To demonstrate the fine-grained analysis possible with our Bayesian algorithm, we also quantified the variability in rule-switch responses across rats by calculating for each rat the proportion of trials in which each rule strategy was above chance after the switch (*Figure 4c, d*). Individual rats sustained their use of the now-incorrect strategy after both spatial → cued and cued → spatial rule switches (*Figure 4c, d*). The reduction in use of a cued strategy after cued → spatial rule switches was also evidenced by that strategy reverting to chance for a few trials so that the proportion of trials above

chance was less than one (*Figure 4d*, right). Rats in both tasks also showed considerable variation in how often a spatial strategy was above chance after the switch to the spatial rule (*Figure 4c, d*, right), suggesting varying rates of adapting to the rule switch.

Our trial-resolution algorithm has thus provided evidence from both learning (*Figure 3f*) and responding to rule switches (*Figure 4*) that the spatial → cued rule switch is harder for rats to detect than the cued → spatial switch, across two different tasks. Moreover, the behavioural response to the cued → spatial rule switch (*Figure 4a, b*, right panels) shows that, as both were above chance, the newly correct strategy (go spatial) can be acquired even while the now incorrect strategy (go cued) is still actively used.

## Independent changes in lose-shift and win-stay around learning and rule switches

Having used the changes in rule strategies to characterise both learning and the response to rule switches, we then sought to understand the exploratory strategies that were driving these changes.

To probe this, we computed $P(\text{strategy}_i(t)|\text{choices}(1:t))$ for four exploratory strategies based on the task features pertinent to the target rules. For the spatial rules, these were win-stay and lose-shift based on the choice of arm or lever; for example, repeating the choice of left arm or lever after being rewarded for it on the previous trial would be a spatial win-stay. For the cued rules, these were win-stay and lose-shift based on the cue position; for example, repeating the choice of the cued arm or lever after being rewarded for it on the previous trial would be a cued win-stay.

We found strong evidence that rule learning was preceded by a high probability of animals using lose-shift in response to the corresponding feature of the target rule (*Figure 5a, d*, top pair of panels). This elevated use of lose-shift to the target-rule's feature was consistent for both classical and strategy definitions of learning trials and across both the Y-maze (*Figure 5a*) and lever-press (*Figure 5d*) tasks. The sustained probability of using lose-shift was not due to the lack of opportunity to update that strategy: before the identified learning trial animals had not yet reached ceiling performance with the rule, so were still incurring losses (*Figure 5a, d*, bottom panels). This use of lose-shift is consistent with latent learning of the relevant feature of the target rule.

Lose-shift exploration rapidly altered at rule changes too. Immediately after a rule switch, lose-shift in response to the old rule's target feature declined rapidly as animals now persevered in applying the old rule, so frequently re-chose the same option after losing (*Figure 5b, c, e, f*). In the lever-press task, there was a rapid increase of lose-shift in response to the new rule's feature for both switches from spatial to cued rules (*Figure 5e*, right) and vice versa (*Figure 5f*, left), again consistent with latent awareness that the original rule had changed.

For both learning and rule changes, win-stay exploration did not change with lose-shift. During learning, win-stay changed at or around the learning trial in both the Y-maze (*Figure 5a*) and lever-press (*Figure 5d*) tasks. At a rule change, win-stay for the old rule's feature changed faster for the cued → spatial (*Figure 5c, f*, right panels) than the spatial → cued rule switches (*Figure 5b, e*, left panels), underpinning the faster response to the cued → spatial switch (*Figure 4*). We ruled out that this independence of win-stay from lose-shift was because one of those strategies could not be updated, as the rate of correct trials was neither at floor nor ceiling before learning and after rule switches (*Figure 5*: bottom panels in a, c, d, f). Lose-shift and win-stay were thus independently used and changed during exploration around learning and rule changes.

## Latent exploratory strategies disguise the absence of learning in humans

We now turn to demonstrating insights from the inference algorithm on a wider range of tasks. Here, we analyse human performance on a simple stimulus-choice task, with no rule switches, to illustrate how the algorithm performs with only a few trials; in the next section we consider non-human primate performance on a stimulus–action task with stochastic reward.

Human participants performed a two-choice task, similar to that of *Pessiglione et al., 2006*, in which they had to choose between two stimuli that appeared above and below a central fixation point (*Figure 6a*). Which of the pair of stimuli appeared above or below fixation was randomised on each trial, and three pairs of stimuli corresponded to three interleaved trial types. In 'gain' trials, one stimulus increased total reward with a probability of 0.8, the other nothing; in 'loss' trials, one

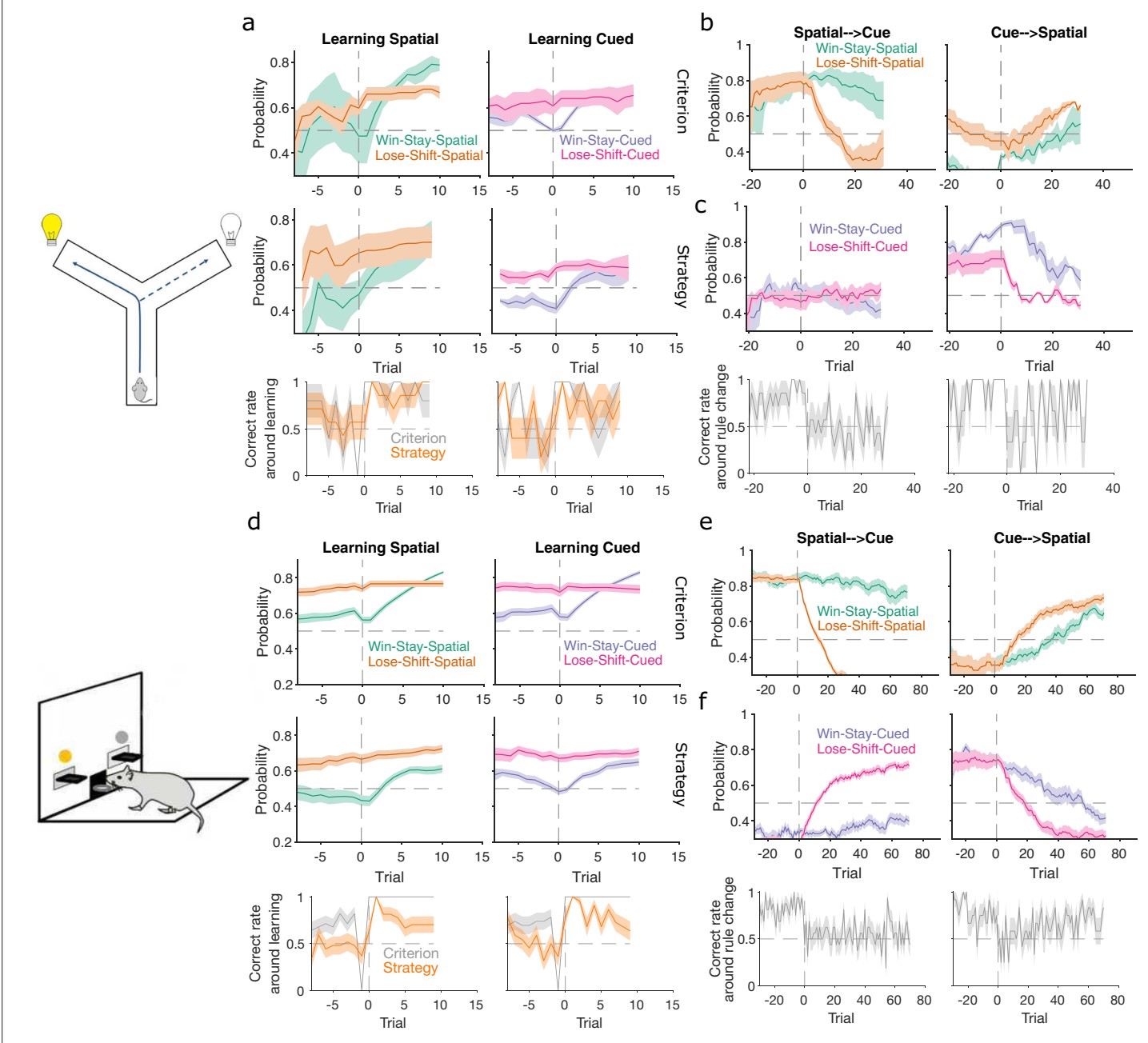

**Figure 5.** Lose-shift and win-stay independently change around learning and rule switches. (**a**) Changes in the probability of win-stay and lose-shift strategies around learning. In the top pair of panels, we plot probabilities for choice-driven (left) and cue-driven (right) forms of win-stay and lose-shift aligned to the identified learning trial (trial 0). Learning trials identified by either the original study's criterion (top) or our rule-strategy criterion (middle). The bottom panel plots the proportion of correct choices across animals on each trial. Lines and shading show means ± standard error of the mean (SEM; $N$ given in **Figure 3b**). (**b**) Rule switch aligned changes in probability for choice-based win-stay and lose-shift strategies ($N$ given in **Figure 4a**). (**c**) Rule switch aligned changes in probability for cue-based win-stay and lose-shift strategies. (**d–f**) Same as panels a–c, but for the lever-press task. Panel d: $N = 28$ rats per curve. Panels e, f: $N = 16$ rats spatial → cued; $N = 12$ cued → spatial.

stimulus reduced total reward with a probability of 0.8, the other nothing; and in 'look' trials, participants randomly chose one of the two stimuli, with no associated outcome. Their implicit goal was thus to maximise reward by learning to select the gain stimulus and avoid the loss stimulus. As participants only experienced 30 of each trial type, these data could present strong challenges to classical approaches of identifying if and when participants learnt to correctly choose the gain stimulus or

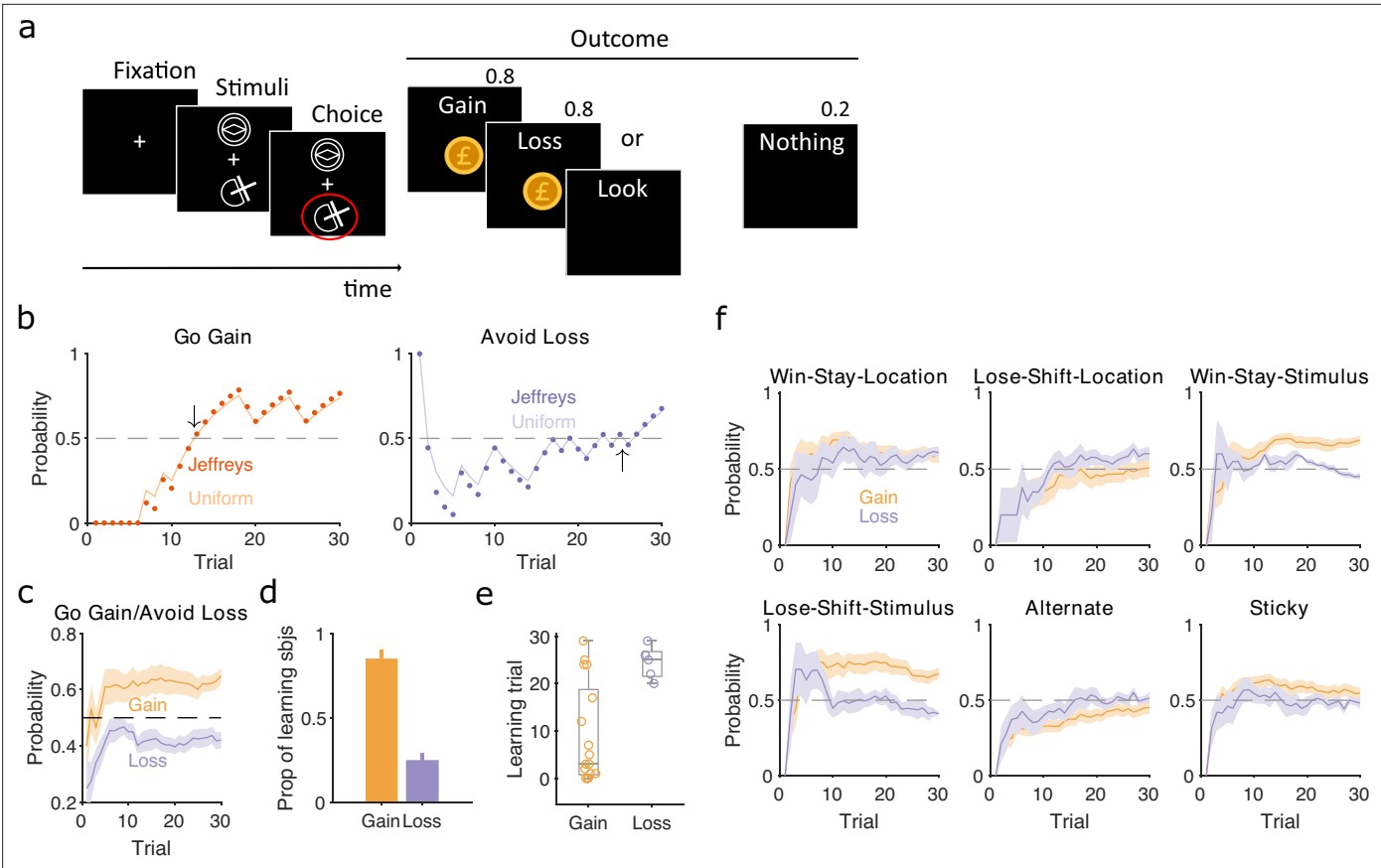

**Figure 6.** Human learning and exploration on a gain/loss task. (**a**) Gain/loss task. Gain, loss, and look trials each used a different pair of stimuli and were interleaved. A pair of stimuli were randomly assigned to the top or bottom position on each trial. Numbers above Outcome panels refer to the probabilities of those outcomes in the gain and loss trial types. (**b**) For an example participant, the maximum a posteriori (MAP) estimate of $P(\text{strategy}_i(t)|\text{choices}(1:t))$ for the target-rule strategy during gain (left panel) and loss (right panel) trials. We plot two estimates, initialised with our default uniform prior (solid) or Jeffrey's prior (dotted). Horizontal dashed lines indicate chance level. Black arrows indicate the learning trial based on our strategy criterion. (**c**) The MAP estimate of $P(\text{strategy}_i(t)|\text{choices}(1:t))$ for the target-rule strategies across all participants. Curves plot mean ± standard error of the mean (SEM) (shading) across 20 participants. The dashed line is chance. (**d**) Proportion of participants that learnt the target strategy for gain or loss trials. Learning was identified by the first trial at which the MAP estimate of $P(\text{strategy}_i(t)|\text{choices}(1:t))$ was consistently above chance until the end of the task (black arrows in panel b). Vertical bar indicates the 95% Clopper–Pearson confidence intervals for binomial estimates. (**e**) Distributions of learning trials for gain and loss trials, for the participants meeting the learning criterion. Each dot is a subject. Boxplots show the median, and the 25th and 75th percentiles. The whiskers extend to the extreme values. (**f**) Probabilities of exploratory strategies for the learning participants (N = 13 for gain, N = 5 for loss). Curves plot mean ± SEM (shaded area) MAP probabilities.

The online version of this article includes the following figure supplement(s) for figure 6:

**Figure supplement 1.** Human strategies are robust to priors and do not correlate with bias in preferred target locations.

avoid the loss stimulus, and what strategies they adopted during learning. In contrast, our Bayesian approach can track changes in strategies within a few trials (***Figure 2***).

To identify learning, we separately computed $P(\text{strategy}_i(t)|\text{choices}(1:t))$ for the strategies of selecting the gain stimulus and of avoiding the loss stimulus (***Figure 6b***). Despite the few trials involved, these estimates were not sensitive to the choice of prior (***Figure 6b*** and ***Figure 6—figure supplement 1***). Across all participants, the MAP estimates of $P(\text{strategy}_i(t)|\text{choices}(1:t))$ showed evident learning of choosing the gain stimulus, but not of avoiding the loss stimulus (***Figure 6c***). Applying the same learning criterion as above, of the MAP probability being above 0.5 until the end of the session, our inference algorithm suggests most participants learnt to choose the gain stimulus (13/20), but not to avoid the loss stimulus (5/20) (***Figure 6d***). Even then, the few who did putatively learn to avoid the loss stimulus did so only towards the end of the 30 trials (***Figure 6e***). Our trial-resolution probabilities have thus shown both strong evidence of which participants learnt the gain trials, and that there is weak evidence at best of any participant learning to avoid the loss stimulus.

To further examine this issue, we asked if the apparent learning to avoid loss could be explained by an exploratory strategy. For the participants who putatively learnt either the gain or loss rule strategies we computed $P(\text{strategy}_i(t)|\text{choices}(1:t))$ for a range of exploratory strategies that captured either responding to the previously chosen stimulus or to the previous location of the chosen stimulus. We found stimulus-driven win-stay and lose-shift were both used for learning gain trials, but not for loss trials (*Figure 6f*). In contrast, only location-driven exploration was consistently, albeit weakly, above chance for loss trials (*Figure 6f*). Because this suggested participants could be sensitive to the location of the stimuli, we then also examined location-dependent strategies in the 'look' trials, and found participants were strongly biased towards choosing one location over the other; however, this bias did not correlate with the participants' use of the avoid loss strategy (*Figure 6—figure supplement 1*). Together, these results suggest that no participant learnt avoid loss, and those who appeared to were instead executing a location-driven exploratory strategy that happened to coincide with avoid loss for the final few trials.

## Choice strategy driven by reward probability not magnitude in a stochastic decision task

More complex decision-making tasks use probabilistic and variable-size rewards for all choices. These present further challenges for the analysis of choice behaviour, not least because the variation of both probability and size of reward increases the number of features upon which subjects may base their choice strategies. To examine these challenges, we used our inference algorithm to analyse a session of data from a rhesus macaque performing a stimulus-primed two-choice task (*Figure 7a*).

In this task, the monkey was presented with a stimulus that predicted the probability of obtaining the larger of two rewards if it subsequently chose either the left or right option on a touchscreen (*Figure 7a*, left). Six stimuli were associated with a different pair of probabilities for obtaining the larger reward for the left or right choice (*Figure 7a*, right); whatever the monkey chose, the small reward was delivered if the probability check for the large reward failed. The stimulus and associated pair probabilities switched once the monkey chose the highest probability option in 80% of trials. With our trial-resolved approach we can immediately ask two questions: did this criterion give sufficient evidence for learning, and was the monkey's choice based on the probability of obtaining the large reward or something else?

To test the switching criterion, we computed $P(\text{strategy}_i(t)|\text{choices}(1:t))$ for the strategy of choosing the side with the highest probability of reward (*Figure 7b*). This confirmed the monkey showed good evidence of learning by the time the criterion was reached, as $P(\text{strategy}_i(t)|\text{choices}(1:t))$ for the highest probability option became dominant at least 10 trials before the switch of reward probabilities (orange line, *Figure 7c*). Our analysis also suggests that in at least one block (trials 25–160 in *Figure 7b*), the monkey had learnt to choose the appropriate option long before the block switch.

As the choice of either option in this task could result in the large reward, the monkey may have been basing its exploration not on the probability of the large reward, but simply on the value of the obtained reward. To examine this, we computed $P(\text{strategy}_i(t)|\text{choices}(1:t))$ for two sets of exploratory strategies that used different features of the monkey's choice history: one set looked at whether repeating the same option choice was driven by the value or the probability of reward on the previous trial (*Figure 7d*); and the other looked at whether switching the option choice was driven by the value or the probability of reward on the previous trial (*Figure 7f*). The monkey's use of exploratory strategies that repeated or switched choices evolved over the blocks (*Figure 7d, f*). Independent of this evolution, both repeating and switching strategies were dominated by the probability of reward rather than the reward obtained (*Figure 7e, g*), evidence that the monkey was basing its decisions more on reward probability than immediate value. Because each session had new arbitrary stimuli for the six pairs of probabilities, the monkey was here re-learning the mapping between stimuli and probabilities, and so was not trivially choosing from an already-learnt association. Rather, these results suggest that, through its prior training, the monkey had learnt the structure of the probabilities, and so was able to use that abstract knowledge to guide its exploratory choices.

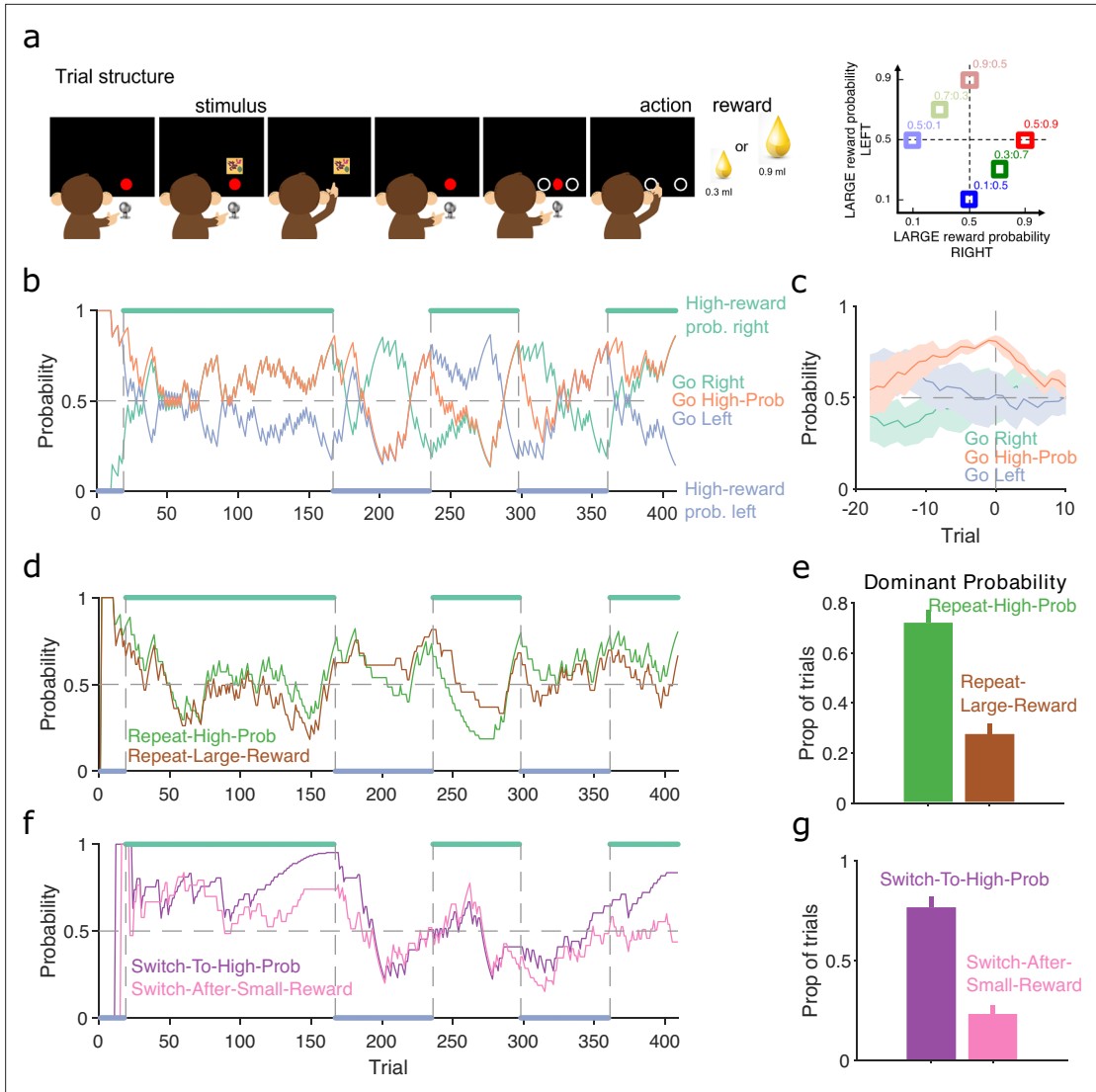

**Figure 7.** Probabilistic reward. (**a**) Schematic of the stimulus-to-action reward task. Left: the monkey initiated a trial by touching a key after the red dot appeared on-screen. A stimulus appeared, predicting which side of the upcoming choice would have the higher probability of obtaining the large reward. Touching the stimulus and then the key brought up the choice options on-screen. The monkey indicated its decision by touching the appropriate circle on-screen. Right: probabilities of obtaining the large reward for each of the six stimuli. (**b**) Maximum a posteriori (MAP) probability for the strategy of choosing the side with the highest probability of reward (go high-probability side). We also plot the MAP probabilities for the choice of side (left or right). Green and blue dots at the top and bottom indicate the side with the higher probability of receiving the large reward; probabilities switched (vertical dashed lines) after the subject chose the high-probability option on 80% of trials in a moving window of 20 trials. (**c**) MAP probabilities in panel b aligned to the transition between blocks (trial 0). Average (bold lines; $n = 5$ blocks) ± standard error of the mean (SEM; shaded areas). (**d**) MAP probabilities along the session for exploratory strategies based on repeating the previous choice after either receiving the large reward or choosing the option with the higher probability of the large reward. (**e**) Proportion of trials in which the two repeating-choice strategies had the largest MAP probability. Vertical bars are 95% Clopper–Pearson confidence intervals. (**f**) As panel d, for exploratory strategies based on switching from the previous choice after either receiving the low reward or choosing the option with the lower probability of high reward. (**g**) As panel e, for the switching strategies in panel f.

## Discussion

Subjects may adopt a range of choice strategies during decision-making tasks, which change during training as they learn a target rule and as the rule is changed. Uncovering how subjects learn to make decisions is essential for understanding what information drives behaviour and for linking neural activity to behaviour. Classical approaches to characterising decision making, such as trials-to-criterion measures (*Birrell and Brown, 2000*; *Peyrache et al., 2009*; *Brady and Floresco, 2015*), psychometric

functions (*Britten et al., 1992*; *Palmer et al., 2005*; *Churchland et al., 2008*), or reward curves (*Gallistel et al., 2004*), cannot track changes over time; algorithms for detecting learning points (*Smith et al., 2004*; *Suzuki and Brown, 2005*) and behavioural change points (*Durstewitz et al., 2010*; *Jang et al., 2015*) are useful, but do not show what strategies were used during learning or around rule changes. To address these issues, we have introduced an approach to decision-making tasks that fully characterises the probability of choice strategies at trial resolution. Our inference algorithm allows tracking both learning, by defining strategies matching target rules, and how subjects learn, by defining exploratory strategies that may or may not depend on features of the task. We have applied our algorithm to a variety of two-choice tasks, across a range of species, including Y-maze and lever-press tasks in rats, a stimulus–outcome mapping task in humans and a stochastic stimulus–action task in non-human primates.

Across the tasks we analysed, our algorithm showed that lose-shift and win-stay are independent exploratory strategies, because a change in the probability of one being executed is not mirrored by a change in the probability of the other. This challenges views that win-stay and lose-shift are complementary strategies, often yoked together (*Miller et al., 2017*; *Izquierdo et al., 2017*; *Aguillon-Rodriguez et al., 2021*; *Roy et al., 2021*).

Emphasising this independence, our analysis shows that changes to lose-shift are evidence of latent learning. Rats adopted a lose-shift strategy matching the target rule before the learning of that rule was identified by overt choice behaviour (*Figure 5a, d*). Conversely, after a switch in rewarded rules in the lever-press task, rats rapidly started using a lose-shift strategy that matched the new rule while still using a win-stay strategy based on the old rule (*Figure 5b, c, e, f*). These data are consistent with ideas that negative and positive feedback can independently drive changes in decision making. Indeed they support the idea that learning what not to do is easier than learning what to do.

We derived three potential criteria for learning from the algorithm's estimate of a subject's probability of using the correct strategy. The most sensitive criterion showed in both the Y-maze and lever-press tasks that learning occurred earlier and more often than detected in traditional approaches based on counting successful trials in some fixed window. All three criteria showed that switching from a spatial to a cue-based rule is harder than the reverse switch. This is consistent with prior work suggesting rats learn spatial rules more easily than visually cued rules (*Mackintosh, 1969*). Prior work on the same lever-press task had shown some evidence of the opposite, that re-learning is faster after switching from a spatial to a cued rule than switching from a cued to a spatial rule (*Floresco et al., 2008*). Our results suggest that the choice of learning criterion do not underpin this discrepancy.

There has been considerable recent advocacy for studying the variety of behaviour within and between individuals (*Krakauer et al., 2017*; *Honegger and de Bivort, 2018*; *Pereira et al., 2020*; *Roy et al., 2021*; *Ashwood et al., 2022*), both to more richly characterise the diversity of behavioural solutions adopted by humans or other animals facing a task, and as a basis for understanding the link between neural activity and behaviour. As it provides trial-resolution probabilities for each subject, our algorithm quantifies the behavioural diversity in a given task. We illustrated that potential here by defining learning points for each subject in the Y-maze (*Figure 3c*), lever-press (*Figure 3f*), and human gain/loss tasks (*Figure 6d, e*), and by showing the distribution of exploratory behaviours around rule switches (*Figure 4c, d*). Further work here could use the trial-resolution probabilities to characterise an individual's repertoire of strategies and when they deployed each strategy.

Reflecting this urgent need for fine-grained behavioural insights into how subjects make decisions, recent work by Pillow and colleagues introduced two innovative solutions to the problem of tracking a subject's behaviour (*Roy et al., 2021*; *Ashwood et al., 2022*). One solution (PsyTrack: *Roy et al., 2021*) continuously infers the parameters of a psychometric function to track the relative influence of different task features on choice in two-alternative forced-choice tasks; the other (*Ashwood et al., 2022*) infers the hidden states underlying decision-making behaviour and the transitions between them, by modelling psychometric functions in each state. Their elegant work applies to well-learnt behaviour, using computationally intensive models fit to all available data post hoc, but lacks the ability to infer when learning occurs. In contrast, our approach focuses on behavioural strategies adopted during learning and rule switches, accumulates evidence up to trial $t$ so can be used on-line, and is able to infer strategies in a few tens of trials (*Figures 2 and 6*), but lacks a model of the transitions between strategies or the parameters which influence the current strategy. An interesting future avenue for our work would be to use the changing probability of different strategies to infer

the transition probabilities between them. Our approach and that of Pillow and colleagues are thus complementary, collectively providing a richer toolbox for understanding a subject's behaviour on choice tasks.

The probabilities computed by our algorithm are a description of the observer: what we can tell about the behaviour on trial $t$ from the evidence so far. This Bayesian analysis of behaviour does not imply the animal is behaving in any optimal, Bayesian way, nor that its brain is computing or using probabilities (*Knill and Pouget, 2004*; *Fiser et al., 2010*; *Pouget et al., 2013*; *Sohn et al., 2019*). Nonetheless, using a Bayesian strategy analysis could let us then formulate those questions of optimality in behaviour and look for neural correlates of probabilistic representations. More generally, such a trial-by-trial analysis of a strategy could then let us look for neural correlates, more accurately tracking the changes in neural activity that precede or align with changes in behavioural strategy (*Rich and Shapiro, 2009*; *Durstewitz et al., 2010*; *Karlsson et al., 2012*; *Powell and Redish, 2016*; *Guise and Shapiro, 2017*).

One issue we faced is that Bayesian inference assumes the parameter being estimated is stationary, but choice strategies are not, so any estimate of their probability must also change. Here, we solve that problem by weighting the evidence entering the Bayesian update by its recency. This allows tracking the probabilities of strategies as they change, and is robust to the exact weighting used (*Figure 2*; *Figure 2—figure supplement 2* to *Figure 2—figure supplement 4*).

The choice of how much to decay the influence of past evidence can determine how quickly a change in strategy is detected (*Figure 2*; *Figure 2—figure supplement 2* to *Figure 2—figure supplement 4*), suggesting other solutions to this non-stationary problem are worth pursuing. For example, having a principled way of varying the weighting $\gamma$ of evidence over time could be useful, potentially so that evidence is rapidly decayed around a switch in strategies and less rapidly during stable execution of a strategy. This would require defining $\gamma$ as a function of some observable parameter of behaviour. It could also potentially find different evidence decay rates $\gamma$ between different strategies.

Introducing $\gamma$ may seem to create a free parameter for the analysis. However, all frequentist approaches to analysing the strategies of subjects also have at least one free parameter, namely the length of the time window over which to compute the probability they are using. This is true both for computing the probability of a particular strategy being used as the proportion of trials it occurs on *Singh et al., 2019*; *Harris et al., 2021*; *Trepka et al., 2021* and for classic learning criterion of subjects performing sequential, or a proportion of, successful trials in some time window (e.g. *Birrell and Brown, 2000*; *Boulougouris et al., 2007*; *Floresco et al., 2008*; *Leeson et al., 2009*; *Peyrache et al., 2009*; *Brady and Floresco, 2015*). Sometimes this time window is explicitly chosen (*Singh et al., 2019*); sometimes the choice is implicit, as the probabilities are computed over trial blocks (*Trepka et al., 2021*) or phases of the experiment (*Harris et al., 2021*). Either way, rarely is the influence of that time window considered. Thus, rather than $\gamma$ introducing a further free parameter, it instead makes the inevitable free parameter of time explicit and bounded: we have placed strong bounds on the values of $\gamma$ by showing the trade-off between the speed of detecting a new strategy and the stability of detecting it.

Our algorithm provides rich information about choice behaviour by computing a full posterior distribution for each modelled strategy, for every trial. How that information is then used to further our understanding of choice behaviour offers a wealth of potential. Here, we tackled the question of when subjects learnt by computing three criteria from the posterior distribution of the target-rule's strategy and the question of how they explored by tracking a point estimate of the probability of each exploratory strategy from their posterior distribution. A third question, which we did not address here, is: which strategy is the subject most likely to be using now? In our simulations, the simple decision rule of choosing the strategy with the highest MAP estimate and highest precision worked well; further work could explore solutions that make fuller use of the posterior distribution, for example computing the probability (*Cinotti and Humphries, 2022*) that a strategy is dominant over the others. Whether this is a good question to ask depends upon our model for how a subject is using strategies: seeking the dominant strategy implies a model where subjects switch between strategies discretely; by contrast, a model where subjects sample from a repertoire of strategies on each trial means our estimated $P(\text{strategy}_i(t)|\text{choices}(1:t))$ is tracking the probability of sampling each strategy. Disentangling these models would give deeper insight into how subjects explore while learning.

The richness of the information depends in part on the choice of strategies examined, which is down to the end-user and what questions they wish to ask of their data. We here distinguished rule strategies and exploratory strategies: while the former should be examined to understand learning, the latter can include a wider range of options than we examined here. We focused predominantly on variants of win-stay and lose-shift strategies as these are widely studied (*Miller et al., 2017*; *Constantinople et al., 2019*; *Aguillon-Rodriguez et al., 2021*; *Roy et al., 2021*; *Ashwood et al., 2022*), contrasting the different features of a task – subject's spatial choice or the task cues – that define the stay or shift response. But the range of strategy models usable with our inference algorithm is broad enough to cover many potential questions about decision making: all the algorithm needs is the success or failure to execute a given strategy on trial $t$. As such, our approach is highly extensible. It can be used with more complex models of choice-history strategies than those considered here: for example, where classic win-stay and lose-shift consider only the previous trial, $n$-back history strategies (*Lau and Glimcher, 2005*; *Akrami et al., 2018*) could capture the effect of choices or outcomes $t − n$ trials ago on the decision at trial $t$. It could also be used with economic games, tracking the probability of each player using specific response strategies to others' decisions, like Tit-for-Tat (*Axelrod and Hamilton, 1981*; *Axelrod and Dion, 1988*; *Nowak and Sigmund, 1993*; *Ohtsuki et al., 2006*). Pilot or bias tests could be used to estimate the prior distribution's parameters for each subject. And while all tasks here had two choices, the algorithm can just as easily compute probabilities of choice strategies in tasks with three or more choices. Despite this extensibility, the computational cost per trial is extremely low, as elementary arithmetic alone is needed for computing the update to the posterior distribution. Consequently, the inference algorithm we introduce here can assess the probabilities of an extensive, flexible range of choice strategies, and could do so in real time, allowing detailed interrogation and potentially closed-loop control of animal and human behaviour during decision-making tasks.

## Materials and methods

### Code availability

Code toolboxes implementing the algorithm and strategy models are available for MATLAB (GitHub) and Python (GitHub). These toolboxes will be updated with future improvements to the algorithm.

All code used to produce the figures in this paper is available at GithHub (copy archived at *Humphries-Lab, 2024*). This repository also includes all data in the paper (see Data Availability Statement).

### Bayesian inference of strategy probability

Here, we give the full algorithm as used throughout the paper.

1. Choose a set of $N$ strategy models to test.
2. Each $P(\text{strategy}_i(t)|\text{choices}(1:t))$ is parameterised by a Beta distribution, with parameters $\alpha_i(t), \beta_i(t)$, for $i = 1, 2, \ldots, N$.
3. The prior is defined by the choice of $\alpha_i(0), \beta_i(0)$. We use the uniform prior $\alpha_i(0) = \beta_i(0) = 1$ by default; we use the Jeffrey's prior $\alpha_i(0) = \beta_i(0) = 1/2$ where noted.
4. Initialise the running counts of successful ($s_i^*$) and failed ($f_i^*$) trials to use strategy $i$ as $s_i^*(0) = f_i^*(0) = 0$.
5. On every trial $t$:
   a. Label it for strategy $i$ as either a:
      Success trial: choice made on trial $t$ is consistent with strategy $i$ $x_i(t) = 1$;
      Failure trial: choice made on trial $t$ is inconsistent with strategy $i$, $x_i(t) = 0$,
      Null trial: strategy $i$ could not be assessed on trial $t$.
   b. For success and failure trials, we update the parameters $\alpha_i(t), \beta_i(t)$:
      i. decay the event totals, and add the new event: $s_i^*(t) = \gamma s_i^*(t − 1) + x(t)$; and $f_i^*(t) = \gamma f_i^*(t − 1) + (1 − x(t))$; with a decay rate $\gamma \in (0, 1]$ .
      ii. update the Beta distribution parameters: $\alpha_i(t) \leftarrow \alpha_i(0) + s_i^*(t)$ and $\beta_i(t) \leftarrow \beta_i(0) + f_i^*(t)$.
   c. For null trials, we assign the previous trial's parameter value: $\alpha_i(t) \leftarrow \alpha_i(t − 1), \beta_i(t) \leftarrow \beta_i(t − 1)$.
6. Output: time-series of $\alpha_i(t)$ and $\beta_i(t)$ for each strategy $i$, which define a Beta distribution for each trial.

Null trials occur when a strategy depends on conditions that could not be met in that trial. In our current implementation we interpolate the previous values for $\alpha, \beta$ for a null trial, to define the probability at which that strategy could have occurred. Future extensions to our work could usefully explore alternative solutions to the handling of null trials.

## Task descriptions

### Y-maze – rats

Rat behavioural data on the Y-maze task data came from the study of *Peyrache et al., 2009*. This dataset includes 50 sessions of four male Long-Evans rats performing a cross-modal rule-switch task. Rats were trained to self-initiate the trial by running up the central stem of the maze and choosing one of the two arms. The rats had to learn in sequence one of four rules: 'go right', 'go cued', 'go left', and 'go uncued'. The cue was a light stimulus randomly switched at the end of one of the two arms. The rule was changed within session after performing 10 consecutive trials or 11 out of 12 correct trials. In the original study, learning was defined as the first of three consecutive trials followed by a performance of ≥80% until the end of the session. For each trial, we used the choice location (right or left), the cue location (right or left), and the reward (yes or no) as input data for the Bayesian model.

### Lever press – rats

All experimental procedures were conducted in accordance with the requirements of the United Kingdom (UK) Animals (Scientific Procedures) Act 1986, approved by the University of Nottingham's Animal Welfare and Ethical Review Board (AWERB) and run under the authority of Home Office project license 30/3357.

Rat behavioural data for the lever-press task were taken from 32 male Lister Hooded rats (aged 11 weeks at the start of testing and food-restricted throughout behavioural training), which served as control groups in two micro-infusion studies (each $N = 16$) targeting prelimbic medial prefrontal cortex. These rats had infusion guide cannulae implanted in the medial prefrontal cortex and received saline infusions in the medial prefrontal cortex (for surgery and infusion procedures, see *Pezze et al., 2014*) at various task stages, including before the sessions during which the data presented in this paper were collected. The task was run in standard two-lever operant boxes with cue lights above the levers and a food well between them (Med Associates, US) based on a previously published training protocol (*Brady and Floresco, 2015*).

Rats were pre-trained to press the levers for sucrose rewards. The rats then had to learn an initial rule to receive one reward per correct response: either a spatial rule or a visual cue-based rule. The spatial rule required the rat to select the left or right lever, one of which was designated as the correct response and associated with reward. The visual cue-based rule required the rat to select the lever indicated by a cue light, which appeared above one of the two levers and was always associated with reward irrespective of which lever it appeared above. Each trial ended either when a response was made on a lever or when the 10-s time window for responses had elapsed without a response (defined as an omission). After reaching the criterion of 10 consecutive correct trials, the rats had to learn to switch their responses to the second rule: rats which had learnt the spatial rule first were tested on the cue-based rule, and vice versa.

Half the rats ($N = 16$) experienced the spatial rule first: all reached criterion on it and on the subsequent cued rule. Half ($N = 16$) experienced the cued rule first: 12 reached criterion so were switched to the spatial rule, which all 12 reached criterion on as well. Our strategy-based definition of learning (see below) found the same number of learners in each condition, so we have $N = 28$ rats throughout for analysis of learning (*Figure 3*) and strategy flexibility (*Figure 4*).

### Gain/loss task – humans

The study was approved by Research Ethics Committee (Stanmore London REC 17/LO/0577). Human participants ($n = 20$, 10 male, 10 female) read a participation information leaflet and undertook informed consent. Participants performed a two choice task, similar to that of *Pessiglione et al., 2006*, in which they had to choose between two arbitrary stimuli that appeared above and below a central fixation point (*Figure 6a*), across three pairs of stimuli corresponding to three interleaved trial types. The top or bottom position of each pair of stimuli was randomised on each trial. In 'gain' trials, one stimulus increased total reward by £1 with a probability of 0.8, the other nothing; in 'loss' trials,

**Table 1.** Rule strategy models for rat tasks.

Columns give the conditions to define trial $t$ as a success, failure, or null for each strategy corresponding to a target rule. The rule strategies we consider here do not have null conditions.

| | Success | Failure | Null |
|---|---|---|---|
| Go left | Chose left option | Did not choose left option | n/a |
| Go right | Chose the right-hand option | Did not choose the right-hand option | n/a |
| Go cued | Chose cued option (e.g. the lit lever) | Did not choose the cued option | n/a |
| Go uncued | Chose the uncued option (e.g. the unlit lever) | Did not choose the uncued option | n/a |

one stimulus reduced total reward by £1 with a probability of 0.8, the other nothing; and in 'look' trials, participants randomly chose one of the two stimuli, with no associated outcome. Thirty trials of each type were presented, randomly interleaved. All participants were informed that the amount they received would be dependent upon their performance, but all received the same amount (£20) at the end of the task.

## Stimulus–action task – non-human primates

Rhesus macaques were trained to touch a key in front of a touchscreen monitor when a red dot appeared on the screen (**Figure 7a**). A visual stimulus then appeared above the dot. After a delay the dot disappeared and the animal had to touch the monitor where the stimulus was present. The dot then reappeared instructing the animal to touch the key again. Two white circles then appeared simultaneously to the left and right of the dot. After a delay the dot disappeared, and the animal chose to select the left or right circle. A large (0.9 ml) or small (0.3 ml) reward of banana smoothie was delivered after every trial. The rewards were probabilistic and asymmetric to either side (**Figure 7a**, right panel). There were six reward probability pairings associated with six visual stimuli per session. For example, one visual stimulus predicted the large reward with p = 0.5 following a left action and p = 0.1 following a right action (denoted '0.5:0.1' in **Figure 7a**). If the probability check for the large reward failed, the small reward was delivered instead. Each of the six stimuli was presented in a blocked design with the stimulus change and block transition occurring after 80% choices (16/20 trials) of the side with the highest probability of the large reward. We analysed here a single session from one monkey.

In the full study from which this session was taken, behavioural data were recorded in two young adult male rhesus macaque monkeys (*Macaca mulatta*, one aged 8 years, weight 10–13 kg; the other aged 9 years, weight 11–15 kg). All animals in the Buckley lab are socially housed (or socially housed

**Table 2.** Exploratory strategy models for rat tasks.

Columns give the conditions to define trial $t$ as a success, failure, or null for each exploratory strategy.

| | Success | Failure | Null |
|---|---|---|---|
| Win-stay-spatial | Rewarded on trial $t-1$ AND chose the same spatial option (e.g. the left lever) on trial $t$ | Rewarded on trial $t-1$ AND NOT chosen the same spatial option on trial $t$ | Unrewarded on trial $t-1$ |
| Lose-shift-spatial | Unrewarded on trial $t-1$ AND chose a different spatial option (e.g. the left lever) on trial $t$ | Unrewarded on trial $t-1$ AND NOT chosen a different spatial option on trial $t$ | Rewarded on trial $t-1$ |
| Win-stay-cue | Rewarded on trial $t-1$ AND chose the same cued option on trial $t$ (e.g. chose the lit lever on trials $t-1$ and $t$; or the unlit lever on trials $t-1$ and $t$) | Rewarded on trial $t-1$ AND NOT chosen the same cued option on trial $t$ | Unrewarded on trial $t-1$ |
| Lose-shift-cue | Unrewarded on trial $t-1$ AND chose a different cued option on trial $t$ from the choice on trial $t$ (e.g. chose the lit lever on trial $t-1$ and the unlit lever on trial $t$; or the unlit lever on trial $t-1$ and the lit lever on trial $t$) | Unrewarded on trial $t-1$ AND NOT chosen a different cued option on trial $t$ | Rewarded on trial $t-1$ |
| Alternate | Chose another spatial option compared to the previous trial | Chose the same spatial option as the previous trial | n/a |
| Sticky | Chose the same spatial option as the previous trial | Chose another spatial option compared to the previous trial | n/a |

**Table 3.** Exploratory strategy models for the non-human primate stimulus-to-action task.
Columns give the conditions to define trial $t$ as a success, failure, or null for each exploratory strategy.

| | Success | Failure | Null |
|---|---|---|---|
| Repeat-Large-Reward | Obtained the large reward on trial $t-1$ AND chose the same option on trial $t$ | Obtained the large reward on trial $t-1$ AND NOT chosen the same option on trial $t$ | Obtained the small reward on trial $t-1$ |
| Repeat-High-Prob | Chose the higher probability of large reward option on trial $t-1$ AND chose the same option on trial $t$ | Chose the higher probability of large reward option on trial $t-1$ AND NOT chosen the same option on trial $t$ | Chose the lower probability of large reward option on trial $t-1$ |
| Switch-After-Small-Reward | Obtained the small reward on trial $t-1$ AND NOT chosen the same option on trial $t$ | Obtained the small reward on trial $t-1$ AND chosen the same option on trial $t$ | Obtained the large reward on trial $t-1$ |
| Switch-To-High-Prob | Chose the lower probability of large reward option on trial $t-1$ AND NOT chosen the same option on trial $t$ | Chose the lower probability of large reward option on trial $t-1$ AND chose the same option on trial $t$ | Chose the higher probability of large reward option on trial $t-1$ |

for as long as possible if later precluded, for example, by repeated fighting with cage-mates despite multiple regrouping attempts) and all are housed in enriched environments (e.g. swings and ropes and objects, all within large pens with multiple wooden ledges at many levels) with a 12-hr light/dark cycle. The NHPs always had ad libitum water access 7 days/week. Most of their daily food ration of wet mash and fruit and nuts and other treats were delivered in the automated testing/lunch-box at the end of each behavioural session (this provided 'jack-pot' motivation for quickly completing successful session performance; supplemented by the trial-by-trial rewards for correct choices in the form of drops of smoothie delivered via a sipping tube) and this was supplemented with fruit and foraging mix in the home enclosure. For the study the animal was prepared for parallel neurophysiological investigation, and so was head-fixated during the task using a standard titanium head-post (and also had recording implants); implantation adopted standard aseptic surgical methods used in the non-human primate laboratory that are not directly relevant for this behavioural study but are described in detail elsewhere (*Wu et al., 2021*). All animal training and experimental procedures were performed in accordance with the guidelines of the UK Animals (Scientific Procedures) Act of 1986, licensed by the UK Home Office, and approved by Oxford's Committee on Animal Care and Ethical Review.

## Strategy models for each task

The strategy models we use for the analysis of the Y-maze and lever-press task data are defined in *Tables 1 and 2*. In *Figures 3–5*, we group the analyses of either 'go left' or 'go right' as 'spatial rules', and the analyses of either 'go cued' or 'go uncued' as 'cued' rules.

Analyses of the human gain/loss task used the same set of exploratory strategies, with the replacement of 'spatial' by 'location', and 'cued' by 'stimulus'. The two rule strategies were defined as: 'go gain' – success if the subject chose the potentially rewarded stimulus; 'avoid loss' – success if the subject did not choose the potentially reward-decreasing stimulus.

Analyses of the non-human primate stimulus–action task examined three rule strategies, 'go left', 'go right', and 'go high-probability', the latter defined as a success if the subject selected the option with the higher probability of the large reward. *Table 3* defines the exploratory strategy models we used for this task.

## Interpreting the posterior distributions

The full posterior distribution $P(\text{strategy}_i(t)|\text{choices}(1:t))$ is a Beta distribution with parameters $\alpha_i(t), \beta_i(t)$. In this paper, we extract two scalar estimates from that posterior distribution in order to visualise changes in strategies and compare strategies.

First, the best point estimate of probability of $P(\text{strategy}_i(t)|\text{choices}(1:t))$ is its mode, also called its MAP value. We find the MAP value by numerically evaluating the Beta distribution's probability density function, and finding the value of $p$ that corresponds to the peak of the density function.

Second, as a summary of evidence for that MAP probability estimate we compute the precision of the posterior distribution, which is the inverse of the variance of the Beta distribution:

$$\text{Precision} = \left( \frac{\alpha\beta}{(\alpha + \beta)^2(\alpha + \beta + 1)} \right)^{-1}. \tag{2}$$

### Learning criteria

We tested three different criteria for defining learning based on the posterior distribution of $P(\text{strategy}_i(t)|\text{choices}(1:t))$ for the target rule.

### Sequence criterion

The first trial at which the MAP probability estimate for the target-rule's strategy remained above chance until the end of the session. If the first trial of a session met this criterion, so all trials were above chance, we chose the trial with the minimum MAP probability estimate, to identify the trial at which the strategy probability increased thereafter.

### Sequence and precision criteria

An extension of the sequence criterion to account for the evidence for each strategy: the trial at which both the MAP probability estimate for the target-rule's strategy remained above chance until the end of the session and the precision of the target-rule's strategy was greater than for all strategies for the other tested rules.

### Expert criteria

The first trial at which the probability that the posterior distribution $P(\text{strategy}_i(t)|\text{choices}(1:t))$ contained chance fell below some threshold $\theta$, and remained so until the end of the session. We use $\theta = 0.05$ here.

## Performance simulations

### Synthetic agent on a 2AFC task

To illustrate the Bayesian algorithm, we generated synthetic data that simulated an agent working on a two-alternative forced-choice task, described in *Figure 2*. The task structure was similar to the tasks described in *Figures 1 and 3*. The agent could choose between two options (e.g. a right or left lever), and a random cue was alternated between the two options. A reward was delivered only when the agent's choice matched the cue. We generated 500 trials divided into blocks of 100 trials, in which the agent's choice followed a specific strategy ('Go right', 'Alternate', 'Lose-Shift-Cued', 'Go cued', and 'Lose-Shift-Choice'). We first defined the vector of cue locations as a random binary vector with p = 0.5; then we defined the vector of choices. For 'Go right' and 'Alternate' the agent's choice was consistently right or alternating, respectively. For 'Lose-Shift' strategies, we generated a random vector of binary choices; then, for every trial $t$ corresponding to a loss (i.e. not matching the cue), we changed the choice in trial $t + 1$ in order to match the cue (Lose-Shift-Cued) or choice location (Lose-Shift-Choice). The 'Go cued' block consisted of choices matching the cue location.

To select the most likely strategy being executed at trial $t$, we chose strategy $i$ with the highest MAP estimate of $P(\text{strategy}_i(t)|\text{choices}(1:t))$ resolving ties by choosing the strategy with the maximum precision.

### Switching simulations

To explore the performance of the Bayesian algorithm, we ran extensive simulations of the algorithm detecting the switch between two arbitrary strategies. The general model for the switch between the two strategies generated a time-series of the success ($x_i(t) = 1$) or failure ($x_i(t) = 0$) to execute strategy $i \in \{1, 2\}$ on each trial. Strategy 1 was executed on trials 1 to $T$ with probability $p_1 = a$, and strategy 2 was executed on trials 1 to $T$ with probability $p_2 = b$. From trial $T$ the probabilities then reversed linearly:

$$p_1 = \begin{cases} a - m \times j & \text{if } j < (a - b)/m \\ b & \text{otherwise} \end{cases}$$

$$p_2 = \begin{cases} b + m \times j & \text{if } j < (a - b)/m \\ a & \text{otherwise} \end{cases}$$

where $j$ is the number of trials elapsed since $T$, and $m$ is the rate of change in probability per trial. *Figure 2—figure supplement 2a* illustrates the model for the case of $a = 1$ and $b = 0$. Note that the probabilities crossover at $j = (a - b)/2m$; we use this in our definitions of detecting the switch below. From this general model we could explore qualitatively different types of strategy switch: whether the switch was abrupt ($m$ is infinity) or gradual ($m < a - b$).

Given some choice of $\{a,b,T,m\}$, we generated the time-series of $x_1$ and $x_2$ for strategies 1 and 2. On every trial $t$ we applied our algorithm to update the two posterior distributions for strategies 1 and 2, and their corresponding MAP estimates $\hat{p}(\text{strategy}_1)$ and $\hat{p}(\text{strategy}_2)$. We defined the detection of the switch as the first trial for which $\hat{p}(\text{strategy}_2) \geq \hat{p}(\text{strategy}_1)$. For abrupt-switching models this was counted from trial $T$; for gradual-switching models this was counted from the 'crossover' trial at which $p_2$ first exceeded $p_1$. Stability of detection was defined as the proportion of the next $w$ trials for which this detection criterion remained true; results are presented for $w = 50$.

As shown in *Figure 2f*, using $\gamma < 1$ causes the detection time to plateau with increasing $T$, because evidence decay means past history has no effect on detection time. For further simulations (*Figure 2—figure supplement 2*) we thus fixed $T = 500$, to get worst-case performance for detection.

For the detection of a new conditional strategy, we defined a probability per trial $p(\text{condition met}) \in (0, 1]$ that its condition would be met and simulated a model where its probability of execution $p_{\text{new}}(t)$ increased linearly from 0 to 1 at a rate of $m$ per trial. We then created time-series of $x_{\text{new}}(t)$, by first checking if its condition was met, setting $x_{\text{new}}(t) = $ null if not; if it was met, we then checked if the strategy was executed ($x_{\text{new}}(t) = 1$) or not ($x_{\text{new}}(t) = 0$). We used our algorithm to find the corresponding MAP estimate $\hat{p}(\text{strategy}_{\text{new}})$ from the time-series, and defined detection as the MAP estimate being greater than 0.5. Results are plotted in *Figure 2—figure supplement 4*.

## Effects of missing the true exploratory strategy

A set of tested exploratory strategies may not contain the true exploratory strategy or strategies used by a subject. To examine how the algorithm performs in this case, we determined the proportion of trials on which a given tested strategy would be expected to make the same choice as a true strategy. Formally, this is the conditional probability $p(x_{\text{test}} = 1|\text{true strategy})$ that the tested strategy is a success on any trial given that the true strategy is being used, which we call $p(\text{match})$ in the main text. The algorithm's MAP probability for the tested strategy is an estimate of this conditional probability.

To compute $p(x_{\text{test}} = 1|\text{true strategy})$, we decomposed this conditional probability into two main terms, one for trials where the true strategy can be executed, and one for trials where it cannot:

$$p(x_{\text{test}} = 1|\text{true strategy}) = p(x_{\text{test}} = 1|x_{\text{true}} = 1)p(x_{\text{true}} = 1) + p(x_{\text{test}} = 1|x_{\text{true}} = \text{null})p(x_{\text{true}} = \text{null}), \quad (3)$$

with $p(x_{\text{true}} = 1) + p(x_{\text{true}} = \text{null}) = 1$. To understand what happens if the true strategy is missing, we enumerated the possible values of *Equation 3* for four cases: both true and tested strategies were unconditional; one was unconditional and the other conditional (giving two cases); and both were conditional. We explain the enumerations for these cases below, and plot the results in *Figure 2—figure supplement 5*.

We first considered the case where the true and tested exploratory strategies were unconditional. In this case $p(x_{\text{true}} = 1) = 1$, and so we needed only to consider the possible values of $p(x_{\text{test}} = 1|x_{\text{true}} = 1)$. For a task with $n$ choices these are $\{0, 1/n, 1 - 1/n, 1\}$: 0, if the tested and true strategy are mutually exclusive (e.g. one repeats the previous choice and one makes a different choice to the previous one); $1/n$, if the true and tested strategy make a single choice; $1 - 1/n$ if the true and tested strategy make anything other than a single choice (e.g. the shift to any other choice than the previous one); 1, if the tested and true strategy are equivalent. The latter implies the true strategy is not missing, so we omitted it; consequently, for the case where both true and test strategies are unconditional, *Equation 3* gives just three solutions $p(x_{\text{test}} = 1|\text{true strategy}) = \{0, 1/n, 1 - 1/n\}$; we plot these three solution as a

cumulative distribution in *Figure 2—figure supplement 5*, panel a. The maximum MAP estimate of $P(\text{strategy}_i(t)|\text{choices}(1:t))$ is thus $1 - 1/n$ for this case.

For the case where the true strategy is unconditional and the tested strategy is conditional, we first decomposed the probability $p(x_{\text{test}} = 1|x_{\text{true}} = 1)$ into the probability that the condition for the test strategy is met $p(\text{test condition met})$ and the probability that the test strategy's choice, if made, matches the true strategy $p(\text{test match true}|\text{condition met})$:

$$p(x_{\text{test}} = 1|x_{\text{true}} = 1) = p(\text{test match true} \mid \text{condition met})p(\text{test condition met}). \tag{4}$$

To evaluate *Equation 3* using *Equation 4*, we tested $p(\text{test condition met}) \in [0.1, 1)$, and used $p(\text{test match true} \mid \text{condition met}) \in \{0, 1/n, 1 - 1/n, 1\}$ as argued for the unconditional case, above. We then evaluated *Equation 3* for all combinations of these values in *Equation 4*, and plot the full set of enumerated probabilities in *Figure 2—figure supplement 5*, panel d, as a cumulative distribution, omitting the case where the tested and true strategies were entirely equivalent (i.e. $p(x_{\text{test}} = 1|x_{\text{true}} = 1) = 1$). Note that we allowed the possibility of the tested strategy being logically equivalent to the true strategy when the tested strategy's condition was met, by including $p(\text{test match true}| \text{condition met}) = 1$; if we do not allow this possibility, we get the set of enumerated probabilities plotted in purple in *Figure 2—figure supplement 5*, panel b.

For the case where the true strategy is conditional and the tested strategy is unconditional, we set $p(\text{true} = 1) \in [0.1, 1)$ to model the frequency at which the true strategy's condition was met. We noted that $p(x_{\text{test}} = 1|x_{\text{true}} = 1)$ and $p(x_{\text{test}} = 1|x_{\text{true}} = \text{null})$ were independent and can each take any value in $\{0, 1/n, 1 - 1/n, 1\}$, except both could not be equal to 1 as this would mean the tested and true strategy were logically equivalent. We then evaluated *Equation 3* using all combinations of these values for $p(\text{true} = 1)$, $p(x_{\text{test}} = 1|x_{\text{true}} = 1)$, and $p(x_{\text{test}} = 1|x_{\text{true}} = \text{null})$, and plot the enumerated probabilities in *Figure 2—figure supplement 5*, panel b.

For the case where both the tested and true strategy are conditional, we again used *Equation 4* to model the conditional test strategy. We then evaluated *Equation 3* using all combinations of values for $p(\text{true} = 1)$ and $p(x_{\text{test}} = 1|x_{\text{true}} = \text{null})$ specified in the preceding paragraph, and for $p(x_{\text{test}} = 1|x_{\text{true}} = 1)$ specified under *Equation 4*. The resulting enumerated probabilities are plotted in *Figure 2—figure supplement 5*, panel e.

In reality we only get to specify the type of test strategy and do not know the type (conditional or unconditional) of true strategy adopted by the subject. Thus, to put bounds on what we can infer from the observed MAP probabilities for a given type of test strategy, we combined the enumerated probabilities for the unconditional and conditional true strategies. The resulting distributions are plotted in the last column of *Figure 2—figure supplement 5*, and from these we derive the three bounds discussed in the main text.

## Derivation of the iterative update and evidence decay

We first rehearse the standard Bayesian approach to estimating a binomial probability $p$ as a random variable when that probability is stationary. We then introduce the decay of evidence to track non-stationary probabilities, and obtain the iterative update rules given in the main text.

Our data are a sequence of Bernoulli trials $\tau = 1, 2, \ldots t$, each of which is either a success $x(\tau) = 1$ or a failure $x(\tau) = 0$ of an event. A running total of the successful events is $s(t) = \sum_{\tau=1}^{t} x(\tau)$. The posterior distribution $\pi(p|s)$ for estimating the probability of that number of successful events is then given by Bayes theorem

$$\pi(p|s) = \frac{P(s|p)\pi(p)}{\int_0^1 P(s|p)\pi(p)dp}, \tag{5}$$

where the denominator is the normalisation constant to obtain a probability distribution function. (Note we write $\pi(p|s)$ as $P(\text{strategy}_i|\text{choices}(1:t))$ in the main text when applied to our specific problem of estimating the probability of strategy $i$).

As noted in the main text, the likelihood $P(s|p)$ is the binomial distribution and we chose the prior $\pi(p)$ to be the Beta distribution with parameters $\alpha, \beta$. Using those in *Equation 5* gives

$$\pi(p|s) = \frac{1}{Z} \underbrace{\left[ \binom{t}{s} p^s (1-p)^{t-s} \right]}_{\text{likelihood}} \underbrace{\left[ \frac{\Gamma(\alpha+\beta)}{\Gamma(\alpha)\Gamma(\beta)} p^{\alpha-1}(1-p)^{\beta-1} \right]}_{\text{prior}},$$ (6)

where $\Gamma$ is the gamma function and $Z$ is the denominator in **Equation 5**.

The solution of **Equation 6** is another Beta distribution:

$$\pi(p|s) = \frac{\Gamma(\alpha+\beta)}{\Gamma(\alpha)\Gamma(\beta)} p^{\alpha+s-1}(1-p)^{\beta+t-s-1}$$ (7)

which is equivalent to simply updating the Beta distribution parameters as $(\alpha(0) + s, \beta(0) + t - s)$ given some starting values of $\alpha(0), \beta(0)$, which define the prior distribution.

Rather than updating after $t$ trials have elapsed, we can instead update every trial by using the posterior of the previous trial $\pi(p|s(t-1))$ as the prior for the update on the current trial $t$. To see this, we can rewrite **Equation 5** to update on every trial $t$ by substituting the Bernoulli distribution $p^x(1-p)^{(1-x)}$ for the binomial distribution, and obtain the iterative equivalent of **Equation 7**:

$$\pi(p|s(t)) = Bp^{\alpha(t-1)+x-1}(1-p)^{\beta(t-1)-x},$$ (8)

where $B$ is the constant $\frac{\Gamma(\alpha(t-1)+\beta(t-1))}{\Gamma(\alpha(t-1))\Gamma(\beta(t-1))}$.

**Equation 8** is equivalent to updating the parameters of the Beta distribution on every trial $t$ as $(\alpha(t-1) + x(t), \beta(t-1) + (1 - x(t)))$, given some prior values for $\alpha(0), \beta(0)$:

$$
\begin{aligned}
\text{Trial 1} \quad & \alpha(1) = \alpha(0) + x(1), & \beta(1) = \beta(0) + (1 - x(1)) \\
\text{Trial 2} \quad & \alpha(2) = \alpha(1) + x(2), & \beta(2) = \beta(1) + (1 - x(2)) \\
& \quad\quad\vdots & \vdots \\
\text{Trial } t \quad & \alpha(t) = \alpha(t-1) + x(t), & \beta(t) = \beta(t-1) + (1 - x(t)).
\end{aligned}
$$ (9)

It is this iterative form that allows computationally efficient real-time updates of the probability of each strategy.

We now turn to our solution to the problem that $p$ is not stationary when estimating the use of strategies. Our idea is to decay the evidence entering the posterior update in **Equation 5**. Consider the following models for the exponential decay of the Bernoulli trial outcomes up to trial $t$, for the successes

$$s^*(t) = \sum_{\tau=1}^{t} \gamma^{t-\tau} x(\tau),$$ (10)

and the failures

$$f^*(t) = \sum_{\tau=1}^{t} \gamma^{t-\tau} (1 - x(\tau)),$$ (11)

where $\gamma \in (0, 1]$ is the rate of evidence decay.

We can substitute $s^*$ for $s$ and $f^*$ for $t - s$ in **Equation 6** and obtain the posterior distribution

$$\pi(p|s^*(t), f^*(t)) = Bp^{\alpha+s^*(t)-1}(1-p)^{\beta+f^*(t)-1},$$ (12)

for the estimate of $p$ on trial $t$ given the decayed evidence up to that trial. Again, this is equivalent to updating the Beta distribution parameters as $(\alpha(0) + s^*(t), \beta(0) + f^*(t))$.

To obtain the trial-by-trial update equivalent to **Equation 12**, we note that the iterative forms of **Equations 10 and 11** are

$$s^*(t) = \gamma s^*(t-1) + x(t),$$
$$f^*(t) = \gamma f^*(t-1) + (1 - x(t)),$$ (13)

starting from $s^*(0) = f^*(0) = 0$.

With these the update of the Beta distribution parameters on trial $t$ becomes

$$\alpha(t) = \alpha(0) + \gamma s^*(t-1) + x(t)$$
$$\beta(t) = \beta(0) + \gamma f^*(t-1) + (1 - x(t)).$$
(14)

Together, *Equations 13 and 14* form the algorithm we use here.

## Limits on the posterior distribution

Any operation altering the evidence entering the likelihood function in *Equation 6* could limit the changes to the resulting posterior distribution compared to the standard Bayesian update. Our decay models (*Equations 10 and 11*) have the advantage that we can define these limits exactly.

*Equations 10 and 11* are geometric series with asymptotic limits for the total successes $s^*(t)$ and failures $f^*(t)$. They thus also set asymptotic limits for $\alpha$ and $\beta$ (*Equation 12*). For a long run of successes where $x(t) = 1$ as $t \to \infty$ these are

$$\alpha \to \alpha(0) + \frac{1}{1-\gamma}, \quad \beta \to \beta(0).$$
(15)

Similarly, for a long run of failures where $x(t) = 0$ as $t \to \infty$ these are

$$\alpha \to \alpha(0), \quad \beta \to \beta(0) + \frac{1}{1-\gamma}.$$
(16)

Consequently, setting the decay parameter $\gamma < 1$ creates asymptotic limits for the values of $\alpha$ and $\beta$ that could limit how much the posterior $P(\text{strategy}_i|\text{choices}(1:t))$ changes from the prior.

We checked the potential limits on the posterior by calculating it at the asymptotic values of $\alpha$ and $\beta$ for long-run successes (*Equation 15*) across the full range of $\gamma \in (0, 1]$. We then calculated the MAP, precision, and the proportion of the posterior that exceeds chance for two options (i.e. p = 0.5). These showed that while the asymptotic limits do not affect the MAP estimate of $P(\text{strategy}_i|\text{choices}(1:t))$, as expected, they do limit how concentrated the posterior becomes around that MAP estimate, which we plot in *Figure 2*.

## Acknowledgements

This work was supported by the Medical Research Council [grant numbers MR/J008648/1, MR/P005659/1, and MR/S025944/1] to MDH and [grant number MR/K005480/1] to MB, and the Biotechnology and Biological Sciences Research Council [grant number BB/T00598X/1] to MB and MDH. SM was supported by an Anne McLaren Fellowship from the University of Nottingham. MS was supported by a Medical Research Council fellowship and a grant from the Dowager Countess Eleanor Peel Trust. Collection of the data on the rat lever-press task was supported by the Biotechnology and Biological Sciences Research Council (BBSRC) Doctoral Training Programme (DTP) at the University of Nottingham [grant number BB/M008770/1, project 1644954], and we are grateful to Stan Floresco for providing MedAssociates programmes to run the lever-press task. We thank Hazem Toutounji for comments on the manuscript, Nathan Lepora for discussions, and Lowri Powell for contributing to the Python port of the toolbox.

## Additional information

### Funding

| Funder | Grant reference number | Author |
| --- | --- | --- |
| Medical Research Council | MR/J008648/1 | Mark D Humphries |
| Medical Research Council | MR/P005659/1 | Mark D Humphries |
| Medical Research Council | MR/S025944/1 | Mark D Humphries |

| Funder | Grant reference number | Author |
|---|---|---|
| Medical Research Council | MR/K005480/1 | Mark Buckley |
| Biotechnology and Biological Sciences Research Council | BB/T00598X/1 | Mark Buckley Mark D Humphries |
| Biotechnology and Biological Sciences Research Council | BB/M008770/1 | Rebecca M Hock Paula M Moran Tobias Bast |
| University of Nottingham | Anne McLaren Fellowship | Silvia Maggi |
| Medical Research Council | | Musa Sami |
| Dowager Countess Eleanor Peel Trust | | Musa Sami |

The funders had no role in study design, data collection, and interpretation, or the decision to submit the work for publication.

## Author contributions

Silvia Maggi, Data curation, Software, Formal analysis, Validation, Visualization, Writing - original draft; Rebecca M Hock, Martin O'Neill, Musa Sami, Investigation, Writing - review and editing; Mark Buckley, Supervision, Funding acquisition, Investigation, Writing - review and editing; Paula M Moran, Tobias Bast, Supervision, Writing - review and editing; Mark D Humphries, Conceptualization, Software, Formal analysis, Supervision, Funding acquisition, Visualization, Methodology, Writing - original draft, Project administration

## Author ORCIDs

Silvia Maggi http://orcid.org/0000-0001-6533-3509
Rebecca M Hock http://orcid.org/0000-0002-0917-570X
Mark Buckley http://orcid.org/0000-0001-7455-8486
Tobias Bast http://orcid.org/0000-0002-6163-3229
Mark D Humphries http://orcid.org/0000-0002-1906-2581

## Ethics

The human gain/loss task study was approved by Research Ethics Committee (Stanmore London REC 17/LO/0577). All participants were read a participation information leaflet and undertook informed consent.
Rat – lever-press task: All experimental procedures were conducted in accordance with the requirements of the United Kingdom (UK) Animals (Scientific Procedures) Act 1986, approved by the University of Nottingham's Animal Welfare and Ethical Review Board (AWERB) and run under the authority of Home Office project license 30/3357. Non-human primate task: All animal training and experimental procedures were performed in accordance with the guidelines of the UK Animals (Scientific Procedures) Act of 1986, licensed by the UK Home Office, and approved by Oxford University's Committee on Animal Care and Ethical Review.

## Decision letter and Author response

Decision letter https://doi.org/10.7554/eLife.86491.sa1
Author response https://doi.org/10.7554/eLife.86491.sa2

# Additional files

## Supplementary files

• MDAR checklist

## Data availability

Source data from the rat Y-maze task data are available from https://crcns.org/ at http://dx.doi.org/10.6080/K0KH0KH5. Source data from the rat lever-press task (32 rats), the human gain/loss task (20 participants), and the primate stimulus-to-action task (one session) are available from the Nottingham Research Data Management Repository at http://doi.org/10.17639/nott.7274. Processed

data and analysis code to replicate all figures are available in our GitHub repository https://github.com/Humphries-Lab/Bayesian_strategy_analysis_Paper (copy archived at *Humphries-Lab, 2024*). Our copies of the source data for the Y-maze task, lever-press task, gain/loss task, and stimulus-to-action task are also freely available from the same repository.

The following dataset was generated:

| Author(s) | Year | Dataset title | Dataset URL | Database and Identifier |
|---|---|---|---|---|
| Hock R, Bast T, Buckley M, O'Neill M, Moran P, Maggi S, Humphries M | 2023 | Experimental data for the trial-resolution strategy inference paper | http://doi.org/10.17639/nott.7274 | Nottingham Research Data Management Repository, 10.17639/nott.7274 |

The following previously published dataset was used:

| Author(s) | Year | Dataset title | Dataset URL | Database and Identifier |
|---|---|---|---|---|
| Peyrache A, Khamassi M, Benchenane K, Wiener SI, Battaglia F | 2018 | Activity of neurons in rat medial prefrontal cortex during learning and sleep | http://doi.org/10.6080/K0KH0KH5 | Collaborative Research in Computational Neuroscience, 10.6080/K0KH0KH5 |

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
