## [Editor Report]

This work describes a valuable method for indexing trial-by-trial learning and decision making strategies in animal and human behavior. The study provides compelling evidence for the validity of this new method.

---

## [Decision Letter]

**Decision letter after peer review:**

Thank you for submitting your article "Tracking subjects' strategies in behavioural choice experiments at trial resolution" for consideration by *eLife*. Your article has been reviewed by 2 peer reviewers, and the evaluation has been overseen by a Reviewing Editor and Joshua Gold as the Senior Editor.

Essential revisions:

1. The method ignores the precision of the posterior in its selection of the best strategy. This is more of a "winner-take-all" approach rather than a method that exploits the Bayesian framework to take the uncertainty of the posterior into account.

2. The authors do not convincingly demonstrate that their method is robust to the presence or absence of a true strategy. Both reviewers ask for an analysis that shows what happens to the prediction when the true strategy is present or absent.

3. Figures 3 c through f should be clarified.

4. The choice of strategies to test the approach is limited. Strategies are static and are unparametrized for choice stochasticity and trial history dependence. Given that more sophisticated strategies are not explored, it is unclear whether this method can be useful in arbitrating between those.

5. Win-Stay Lose-Shift analysis is confounded and provides limited evidence for what can be learned in real data using the method.

*Reviewer #1 (Recommendations for the authors):*

1. The approach is presented as Bayesian, however, that is really a stretch, in ways that matter for the interpretation of the tool. First, the evidence at each trial is taken to be all or nothing (l94-95), rather than integrating over any potential uncertainty. This might make sense for deterministic strategies but it is unclear why this approach is taken for non-deterministic policies. I note that for all real data, assuming deterministic policies is very unrealistic. Second, the full posterior computed by the Bayesian method isn't fully used to compare strategies. There should be a better way to incorporate the variance of the posterior into the decision rule (lines 202-204). The variance shouldn't be completely ignored if the MAP probabilities are different. More generally, the full assumptions of the model should be more carefully described when the approach is presented.

2. The strategy space considered is problematic in a few ways, despite the authors' efforts to include probabilistic strategies.

a. First the strategies considered are all static, despite the emphasis on a dynamic environment. In that sense, it should be made extremely clear that this method is purely descriptive, rather than explanatory. It is not a modeling approach, rather a data analysis approach that might allow researchers to answer questions of the type "by when did the animal reliably express a [follow the light] strategy", for example. It cannot offer insights into how the animal arrives at the strategy or learns it.

b. The possibility that the true strategy might be missing is not sufficiently discussed or analyzed. The authors should show comparisons between the case where the true strategy is known and the case where it is not. For example, what would Figure 2b&c look like if the true strategies are not considered?

c. The authors make strong statements that are not adequately supported. For example, "a MAP probability approaching 1 is strong evidence that the tested strategy is, or equivalent to, the true strategy (lines 271-273)." The paper does not include any analysis that supports this statement.

3. The previous two points may limit the usefulness of the new technique. In practice, animals' policies in dynamic environments are unlikely to be stable and unparameterized; most of the literature instead relies on strategies that include noise parameters, and potentially are dependent on trial history in a parameterized way (e.g., multi-choice stickiness, or reward sensitivity). There is no clear way in which such strategies could be considered by the current method; and by contrast, existing model fitting methods would work well for such strategies, which are also more likely to be relevant.

4. Some analyses lack details in descriptions or analysis in a way that makes them hard to evaluate.

a. Figure 3 c/f: the plots are very difficult to read due to scaling, making the comparison difficult.

b. If the strategy is irrelevant in a given trial, the numbers are not updated (i.e., the trial is neither a match nor a non-match). This complicates the win-stay lose shift result interpretation, as it is likely to be a different proportion of win vs. lose trials. This could lead to the apparent slower decay, which would obviate the conclusion that WS-LS is not a single strategy, but two separate strategies.

c. Figures S5-S6 consider the case of probabilistic strategies; however, the task and strategies considered are insufficiently described. In Figure S5, the authors should show p(match) as a function of MAP probability instead of the other way around, because in practice we are more interested in estimating p(match) with MAP probability.

*Reviewer #2 (Recommendations for the authors):*

I find this approach to be extremely compelling and highly useful. I do have one concern about how robust this approach is to the presence or absence of the real strategy in the candidates that are being tested and updated. This is a slightly different question than what was addressed with the enumeration of possible values of p(match) described on pages 8-9. I would like to see a test that compares the MAP estimates of incorrect strategies A and B when the true strategy C is absent versus present in the strategies being tested.

I congratulate the authors on a well-written and coherent manuscript. I have a few specific suggestions for improving the clarity and readability:

General: check your manuscript for use of the passive voice.

Page 9 line 217: consider re-writing the sentence starting with "We deemed […]".

Figure 2, panels h and i: the caption reads "number of trials after the crossover trial until detection, as a function of how quickly the probabilities […] change per trial". While I understand that the word "quickly" is being used as a substitute for the amount of probability change, it is misleading in that the x-axis of those plots has nothing to do with time.

Figure 3, panels c and f: I found these figures to be especially confusing. It is not clear what each dot represents in each case (in some cases it is trials and in others it is animals). The point the authors are trying to convey is thus not effectively conveyed. For example, the takeaway is that there is a significant difference in the learning trials for each rule between the original and the strategy criterion for the first rule and not the second, aside from the p values, the scatter plots do not serve as effective visualizations of this point.

Page 12 line 366: The section title should read "lose-shift, not win-stay" instead of the current version (it is inverted).

---

## [Author Response]

Essential revisions:1. The method ignores the precision of the posterior in its selection of the best strategy. This is more of a "winner-take-all" approach rather than a method that exploits the Bayesian framework to take the uncertainty of the posterior into account.

As we use selection of the best strategy only once, in our view this refers to a minor aspect of analysing a simulation, and not a critique of our main contributions – the algorithm and its insights on learning. But we agree with the underlying sentiment that more could be done to demonstrate the use of having the full posterior, and expand on this below. We note that our aim was to be careful in the paper to separate the method, the estimation of the trial-resolution posterior of p(strategy), from the use of its output by further processing of the posterior, and this comment refers to the latter, not the former.

We use selection of the best strategy only once when quantifying the performance of the algorithm on the example simulation of a synthetic agent. As implied by the phrase “winner-takes-all”, here we use a simple approach of choosing the strategy with the maximum MAP, and breaking ties by choosing the strategy with the highest precision, indicating lower uncertainty. While we suggest in the text this could be a useful approach to apply to data, we readily acknowledged that richer criteria could be derived from the posterior distributions obtained by the algorithm (lines 602-612 in the original submission). To make these points more clearly, we have redrafted the Results text describing the synthetic agent simulation (191-194) and the Discussion section suggesting further development in the use of the algorithm’s output (lines 881-890).

In the paper we outlined two questions we aimed to tackle given the posterior distribution of each strategy: detecting learning and tracking the use of exploratory strategies. Consequently we do not “detect the best strategy” in any data analyses we present (Figure 3-7 and accompanying supplementary figures). We have redrafted the opening paragraphs of the Results (lines 74-79) and the relevant paragraph in the Discussion (lines 875-880) to further clarify the core questions of the paper.

On the issue of making further use of the uncertainty in the posterior we have done three things:

We have added a paragraph to the Results after we introduce the algorithm (lines 155 – 160) to draw the reader’s attention to how we use the posterior and what more could be done with it.

We have given a fuller account of the behaviour of the posterior as a function of the forgetting rate (γ): the Methods now include a clearly marked section and text discussing this around Equations 15 and 16 (lines 947-962); and Figure 2 now includes panels g-i showing the posterior’s behaviour.

We developed and tested two further criteria for learning that considered the uncertainty in the posterior. These are presented in the Results (lines 312-326); outlined in the Methods (lines 802-808); and the outcomes of using these criteria on the Y-maze and lever-press task are summarised in Figure 3 – Supplemental Figure 1. We find all criteria replicate the result that rats learnt the switch from a cued to a spatial rule considerably faster than switching from a spatial to a cued rule.

2. The authors do not convincingly demonstrate that their method is robust to the presence or absence of a true strategy. Both reviewers ask for an analysis that shows what happens to the prediction when the true strategy is present or absent.

As noted above, the determination of which is the “true” strategy was not our goal, and we do not use our algorithm in this way in the paper. Rather, we focused on tracking the probability of user-specified strategies, so that we can capture the evidence for learning or for the features driving exploratory choice (e.g. are agents responding to losses or wins; are they responding to cues or choice etc).

One reason for this focus is that to our minds detecting a single “true” strategy is ambiguous. For the observer, multiple strategies may be logically equivalent – for example, a subject that consistently chooses “go right” and gets rewarded is also consistently choosing “win-stay” to their choice. Subjects may be actively using more than one strategy – for example, when they switch strategies when learning a new rule after a previously-established one, then both are often expressed on different trials (e.g. Figure 1d and f). Thus our algorithm seeks to track the probability of the use of each strategy, and interpret these as the observer’s estimate of the likelihood of their expression.

We have extensively redrafted the section on analysing the algorithm’s performance (Section “Robust tracking of strategies in synthetic data”) to clarify the goals of that analysis and how each set of simulations or analyses addresses them: (1) to provide the rationale for the use of evidence decay; (2) to show how evidence decay affects the algorithm’s output, and thus provide a basis for defining usable values of that parameter; and (3) to provide aid in interpreting the resulting values of P(strategy). In the context of (3), we have considered what values P(strategy) can take when a true exploratory strategy is missing. We redrafted the text on the analysis of the missing “true strategy” (lines 239-246) to clarify its aims and insights.

In redrafting this section we also addressed individual reviewer requests for more information on the limits of the method (the behaviour of the posterior, noted above, lines 209-217) and concerns about the choice of values for the evidence decay parameter (esp. lines 220-230).

3. Figures 3 c through f should be clarified.

We have done the following to clarify the results:

To panel c: added a histogram of the number of identified learning trials per animalTo panel f: shown all individual animals’ data in light, open symbols; reduced the jitter of the symbols to align them better with the x-axis labels; and overplotted the mean and standard error of the mean as solid symbols to provide a clearer summary of the main results (of earlier detection of learning by strategy-based criterion during the first learnt rule; and of slower learning of a cued rule when it followed a spatial rule).Added ANOVAs to give statistical support to the result that, in the lever-press task, learning the cued rule after a spatial rule is slower than the reverse.

4. The choice of strategies to test the approach is limited. Strategies are static and are unparametrized for choice stochasticity and trial history dependence. Given that more sophisticated strategies are not explored, it is unclear whether this method can be useful in arbitrating between those.

We politely disagree with this statement. We tested 14 different strategies in the paper (Tables 1-3 of the Methods). These include a range of strategies that use trial history dependence (e.g. Figure 7 and Table 3), and the Discussion already touched on ways to extend to other trial history dependence, such as dependence on outcome N trials in the past. We have redrafted the Results text around Figure 7 (lines 459-463) and the Discussion section (lines 602-606) to make this more explicit.

We have clarified that our results already showed the algorithm can track the stochastic use of a strategy (Results lines 223-227, and Figure 2 – Supplemental Figure 3).

5. Win-Stay Lose-Shift analysis is confounded and provides limited evidence for what can be learned in real data using the method.

We agree that we did not explain this well. Indeed in the absence of losses Lose-Shift cannot be updated, and in the absence of wins Win-Shift cannot be updated. However, this analysis focussed on the trials preceding the detected learning trial or the trials following a rule switch. In both cases, there is a mixture of wins and losses, so the probabilities of both Lose-Shift and Win-Stay can be updated. We now show this explicitly by plotting the per-trial rate of correct choices around both learning and rule-shifts in Figure 5. The accompanying text results have been redrafted to acknowledge the need for evidence of losses and wins [lines 371-373 and 385-387]; we have also simplified the text in the Results discussing Figure 5 to further clarify the focus of the analysis on the trials preceding learning and following rule-switches [lines 367 – 389].

Further Revisions

Reviewer 1 suggested we “Define "strategy" and/or give examples in the Introduction because it can be a vague concept to readers from different backgrounds.” The Introduction now gives explicit examples (lines 40-43).

Reviewer 1 commented that “it should be made extremely clear that this method is purely descriptive, rather than explanatory. It is not a modeling approach, rather a data analysis approach”. We are in firm agreement: indeed the Discussion included a dedicated paragraph on this point beginning “The probabilities computed by our algorithm are a description of the observer…”, and throughout the algorithm was described from the perspective of the observer, not the agent. To make this even clearer, we have edited the Results text introducing the method to emphasise that the evidence and computation are from the observer’s point of view (lines 71, 87-91).

We have revised text throughout for clarity.